# Cell type-specific biotin labeling in vivo resolves regional neuronal and astrocyte proteomic differences in mouse brain

Sruti Rayaprolu[1,2], Sara Bitarafan[3], Juliet V. Santiago[1,2], Ranjita Betarbet [1,2], Sydney Sunna[1,2], Lihong Cheng[1,2], Hailian Xiao[1,2], Ruth S. Nelson[1,2], Prateek Kumar[1,2], Pritha Bagchi [2,4,5], Duc M. Duong[2,4,5], Annie M. Goettemoeller[6], Viktor János Oláh[6], Matt Rowan [6], Allan I. Levey [1,2], Levi B. Wood[3], Nicholas T. Seyfried [2,4,5✉] & Srikant Rangaraju [1,2✉]

Proteomic profiling of brain cell types using isolation-based strategies pose limitations in resolving cellular phenotypes representative of their native state. We describe a mouse line for cell type-specific expression of biotin ligase TurboID, for in vivo biotinylation of proteins. Using adenoviral and transgenic approaches to label neurons, we show robust protein biotinylation in neuronal soma and axons throughout the brain, allowing quantitation of over 2000 neuron-derived proteins spanning synaptic proteins, transporters, ion channels and disease-relevant druggable targets. Next, we contrast Camk2a-neuron and Aldh1l1-astrocyte proteomes and identify brain region-specific proteomic differences within both cell types, some of which might potentially underlie the selective vulnerability to neurological diseases. Leveraging the cellular specificity of proteomic labeling, we apply an antibody-based approach to uncover differences in neuron and astrocyte-derived signaling phospho-proteins and cytokines. This approach will facilitate the characterization of cell-type specific proteomes in a diverse number of tissues under both physiological and pathological states.

---

[1] Department of Neurology, Emory University, Atlanta, GA 30322, USA. [2] Center for Neurodegenerative Diseases, Emory University, Atlanta, GA 30322, USA. [3] Georgia W. Woodruff School of Mechanical Engineering, Parker H. Petit Institute for Bioengineering and Bioscience, and Wallace H. Coulter Department of Biomedical Engineering, Georgia Institute of Technology, Atlanta, GA 30322, USA. [4] Emory Integrated Proteomics Core, Emory University, Atlanta, GA 30322, USA. [5] Department of Biochemistry, Emory University, Atlanta, GA 30322, USA. [6] Department of Cell Biology, Emory University, Atlanta, GA 30322, USA. ✉email: nseyfri@emory.edu; srikant.rangaraju@emory.edu

The brain is comprised of distinct cell types including neurons, glia (astrocytes, oligodendrocytes, microglia), and vascular cells (endothelial cells and pericytes). Brain cell types demonstrate complex and nuanced changes in their molecular composition (DNA, mRNA, protein) during physiological processes in development, aging, and in pathological states[1]. Recent advances in bulk and single cell transcriptomic profiling of healthy and pathogenic states of mouse and human brain have provided numerous new insights into diverse molecular signatures adopted by brain cell types[2]. Specifically, they have highlighted several causal neuronal and glial cellular responses in neurological diseases; however, transcriptomic findings only modestly correlate with protein-level (proteomic) changes[3–6]. Proteins also undergo post-translational modifications that impact their function and expression profiles quantitatively, temporally, and spatially; a feature not reflected by transcript abundance[3,5,6]. Therefore, proteomic characterization of distinct brain cell types can provide important insights into cellular mechanisms of development, aging and neuropathology.

Proteomic profiling of brain cell types requires isolation of intact cells from fresh unfrozen brain, a pre-requisite that is not necessary for single-nuclear transcriptomic studies of intact nuclei from frozen brain[7,8]. There are numerous methods by which relatively pure, live cells of interest can be isolated with minimal contamination and high sensitivity. For instance, magnetic activated cell sorting (MACS) and fluorescence activated cell sorting (FACS) have been used to isolate one or more brain cell type for mass spectrometry (MS)-based proteomics[9,10]. However, these methods are not without limitations, including variable cell yield, low protein yields, cellular and acellular contamination, sampling biases, and artefactual cellular activation, which significantly impact the proteomic profiles of these cell types[9]. Furthermore, isolation-based proteomic studies of glia are also susceptible to sampling bias and fail to capture molecular signatures representative of their native state. Several studies have characterized the proteome of synapses from mouse and human brain tissues[11,12]; however, live neurons from fresh brain tissue are particularly difficult to isolate, inherently limiting our ability to understand neuron-specific proteomic changes occurring in vivo.

Recently, in vivo cell type-specific transcriptomic labeling approaches (e.g., RiboTag) were developed to quantify transcripts without the need for cell isolation[13–15]. Analogous to these approaches, in vivo proteomic labeling has been achieved by bioorthogonal non-canonical amino acid tagging (BONCAT) where a Cre-lox genetic strategy leads to conditional expression of the mutant (L274G) methionyl tRNA synthetase (MetRS*) in a specific cell type in the mouse brain[16]. MetRS* incorporates a methionine analog, azidonorleucine (Anl), into newly synthesized proteins in place of methionine in the desired cell type[16–19]. Anl-tagged proteins can then be enriched using azide-alkyne "Click" chemistry followed by MS quantitation[19]. To date, in vivo BONCAT has been successfully applied to characterizing the nascent neuronal proteome and is well-suited to study turnover of newly synthesized proteins, potentially in non-neuronal cells as well[16,17]. Synapse-specific proteomics in vivo has also been achieved using the conditional Cre-lox PSD95 tandem affinity purification (cPSD95-TAP) approach[20].

Proximity-dependent protein labeling approaches using an engineered biotin ligase, such as TurboID, or ascorbate peroxidase, APEX, represent alternative strategies for global or subcellular proteomic labeling within a cell[21]. TurboID can rapidly label proteins in a 9–10 nm radius under physiological conditions making this highly suitable for labeling in live cells in vivo[22]. This allows for the investigation of dynamic biological processes with greater proteomic breadth. In vivo applications of TurboID have been limited to investigating protein-protein interactions by

fusing TurboID to the protein of interest or obtaining peripheral cell type-specific secretome profiles[22]. The proteomic profiles of astrocyte-synapse interactions were characterized in the mouse brain by introducing TurboID via adeno-associated viruses (AAV)[23], but brain cell type-specific proteomics using TurboID has been limited by lack of a suitable mouse model. APEX oxidizes biotin-phenols into biotin-phenoxyl radicals in the presence of hydrogen peroxide ($H_2O_2$) and these short-lived radicals biotinylate proteins within a radius of several nanometers. In the mouse brain, APEX has been used to study the subcellular proteomics of neurons within the striatum as well as dopaminergic neurons[24,25]. Unlike TurboID, APEX-mediated biotinylation is limited to ex vivo applications because $H_2O_2$ is toxic to live animals.

Here, we developed and validated a $Rosa26^{TurboID}$ mouse line for TurboID expression via Cre-lox genetic approaches to induce TurboID expression in specific cell types. We term this approach Cell type-specific In vivo Biotinylation Of Proteins or CIBOP. We first used an AAV mediated delivery (AAV-CIBOP) of Cre recombinase under the human synapsin (hSyn1) promoter for pan-neuronal labeling of proteins in the hippocampus. Our second application involved a transgenic approach (Tg-CIBOP) by crossing the $Rosa26^{TurboID}$ mouse with Camk2a-Cre$^{Ert2}$ mice[26,27] to label Camk2a neurons and with Aldh1l1-Cre$^{Ert2}$ mice[28,29] to label astrocytes in the entire brain in an inducible manner. Neuronal AAV-CIBOP and Tg-CIBOP resulted in robust biotinylation of proteins specifically within neuronal cell bodies and axons in the mouse brain without histological or electrophysiological abnormalities. We successfully enriched these biotinylated proteins from total brain homogenates and captured neuronal and astrocyte proteomes (>2,000 proteins in each cell type) by MS and identified >200 proteins that differentiated neurons and astrocytes. Furthermore, we were able to resolve unique proteomic signatures of Camk2a neurons and Aldh1l1 astrocytes that are brain region-specific and indicative of distinct cellular functions and disease-vulnerability. Next, we complemented our untargeted proteomic approach with a targeted measurement of biotinylated phospho-proteins via immunoassays from key cellular signaling pathways (MAPK and Akt/mTOR) as well as biotinylated cytokines that are often below MS detection limits. This enabled us to quantify neuron-derived signaling phosphoproteins and cytokines, demonstrating regional expression patterns in the brain. Lastly, we extended this approach to contrast signaling phosphoprotein profiles of neurons and astrocytes, revealing relatively increased activation of MAPK signaling in neurons. The $Rosa26^{TurboID}$ mouse, our validated approaches for cell type-specific in vivo proteomic labeling, and optimized workflow for proteomic characterization represent a highly promising approach for global cellular proteomics of desired cell types in their native state in vivo without the need for cell type isolation. Critically, these neuron- and astrocyte-specific proteomic data provide a resource of native-state proteomes of brain cell types.

## Results

**Hippocampal pan-neuronal proteomics using an adeno-associated viral (AAV) strategy.** We first developed and validated a targeting vector designed to insert TurboID into the $Rosa26$ locus in mice and then generated the $Rosa26^{TurboID}$ mouse line (Fig. 1a and Supplementary Fig. 1a–c). In a cohort of 3-month-old $Rosa26^{TurboID/wt}$ mice (Fig. 1a) and wild-type (WT) littermate controls, we stereotaxically injected AAV9 carrying the Cre recombinase gene under the hSyn promoter (AAV9-hSyn-Cre) into bilateral hippocampi for pan-neuronal TurboID expression (Fig. 1b). Un-injected WT mice also served as negative

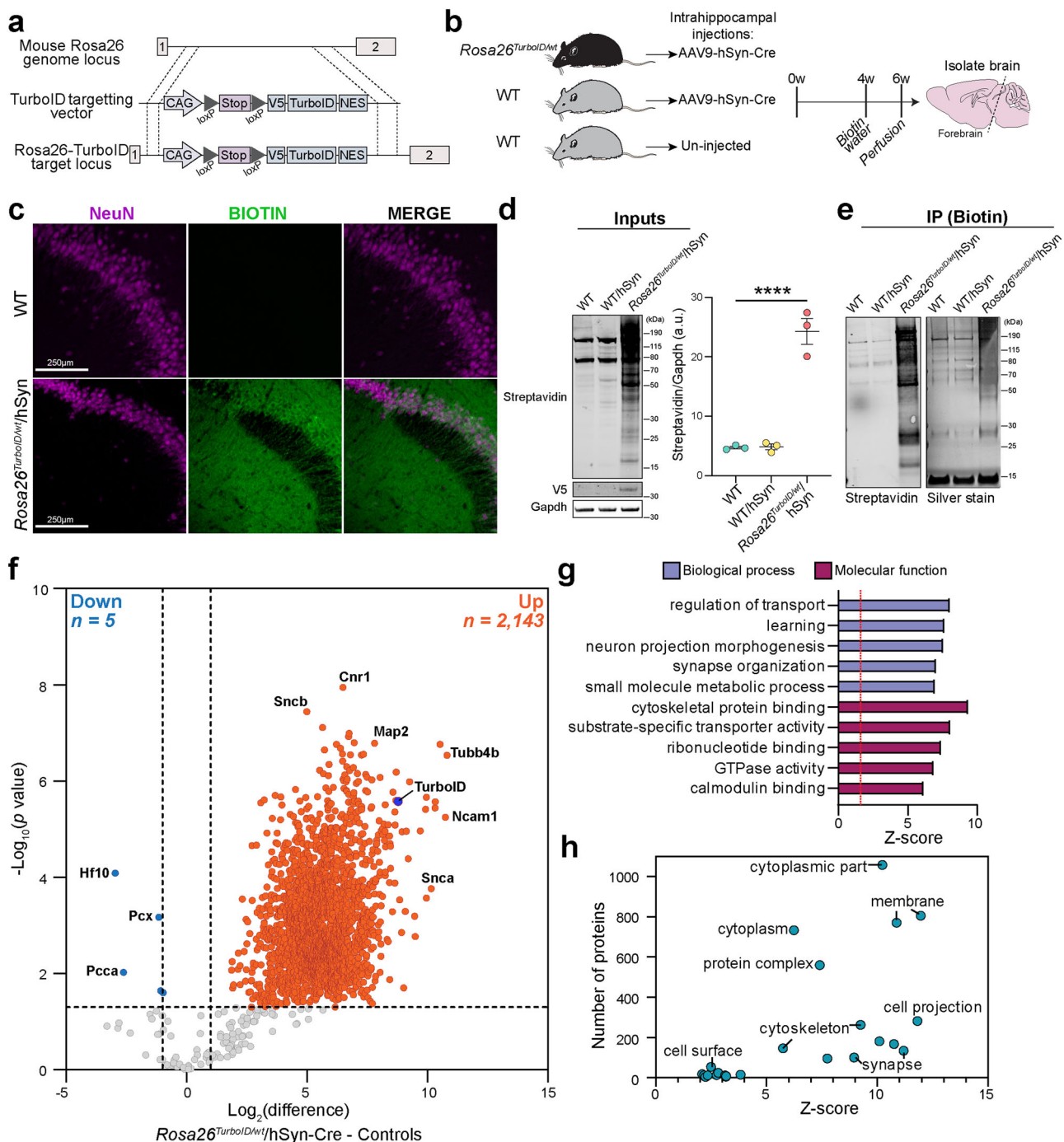

controls (Fig. 1b). Following 4 weeks, mice then received biotin supplementation in water (37.5 mg/L)[30] for 2 weeks with no observable adverse effects (e.g., weight and locomotor activity). Immunofluorescent imaging of hippocampal regions confirmed neuronal (NeuN) biotinylation (detected by streptavidin Alexa 488) in cell bodies, axons, and synapses in majority of hippocampal CA2/3 neurons (Fig. 1c). Western blot analysis of forebrain lysates from mice confirmed TurboID expression (based on V5 detection) and robust biotinylation of proteins in the *Rosa26*[TurboID/wt]/hSyn mice as compared to minimal endogenous biotinylation observed in control mice (Fig. 1d).

Biotinylated proteins were enriched from forebrain lysates using streptavidin beads. Enrichment was confirmed by eluting a small fraction of bound proteins for confirmatory Western blot

and silver stain studies, which showed maximum biotinylated protein yield in the *Rosa26*[TurboID/wt]/hSyn mice (Fig. 1e). After label-free quantitation via mass spectrometry (LFQ-MS) of enriched biotinylated proteins, we quantified 2307 proteins. We first compared *Rosa26*[TurboID/wt]/hSyn mice with all control mice and identified 2,143 proteins with a significant ($p \le 0.05$) $\ge$ 2-fold enrichment, while only 5 proteins with $\ge$2-fold enrichment in the control proteome were identified (Fig. 1f, Supplementary Data 1), which included endogenously biotinylated carboxylases and keratins that were enriched by streptavidin binding. By including phosphorylation of Ser/Thr/Tyr residues in the searches, we identified 147 phospho-peptides from 110 unique proteins of which 55 showed at least 2-fold higher levels in the labeled neuronal proteome (e.g., Map2, Ncam1, Mapt, Map1b, Amph,

**Fig. 1 Development and validation of the *Rosa26*^TurboID/wt^ mouse line for cell type-specific proteomics. a** Genetic strategy for targeting TurboID (V5-TurboID-NES) to the *Rosa26* locus. **b** Schematic of AAV studies to direct Cre recombinase expression in hippocampal neurons ($n = 3$ WT mice with no injections, $n = 3$ WT mice received AAV9-hSyn-Cre, $n = 3$ *Rosa26*^TurboID/wt^ mice received AAV9-hSyn-Cre). **c** Representative immunofluorescence images of CA2/3 of the hippocampus from WT/hSyn ($n = 3$ mice) and *Rosa26*^TurboID/wt^/hSyn ($n = 3$ mice) showing biotinylation (green: streptavidin Alexa488) in relation to neuronal nuclei (magenta: NeuN). **d** Western blot of brain lysates (representative animal from $n = 3$ mice/group) probed with streptavidin fluorophore, anti-V5, and anti-Gapdh antibodies. *Rosa26*^TurboID/wt^/hSyn brain showed biotinylated proteins of different molecular weights as compared to few endogenously biotinylated proteins in the two control groups. Right: Densitometry confirming significant increase in biotinylation signal in *Rosa26*^TurboID/wt^/hSyn brains (one-way ANOVA, ****$p = 0.0001$; data represented as mean ± SEM). **e** Western blot (left) and silver stain (right) of enriched biotinylated proteins after streptavidin-pulldown and release of biotinylated proteins from 10% of streptavidin beads (representative images from $n = 3$ mice/group). As compared to minimal protein enriched in the two control groups, several biotinylated proteins were enriched from *Rosa26*^TurboID/wt^/hSyn brain. **f** Volcano plots showing differentially enriched proteins comparing *Rosa26*^TurboID/wt^/hSyn ($n = 3$) and control mice ($n = 3$ WT and $n = 3$ WT/hSyn). Orange symbols (two-sided *T*-test unadjusted $p \le 0.05$ and ≥2-fold change) represent biotinylated proteins enriched in the *Rosa26*^TurboID/wt^/hSyn brain and examples of neuron-specific proteins are highlighted, in addition to TurboID. Blue symbols represent endogenously biotinylated carboxylases enriched in the control brains. For group wise comparisons, see Supplementary Fig. 1. **g** Results from GSEA of ≥2-fold biotinylated neuronal proteins (orange symbols from panel f), as compared to reference list[5] (mouse brain: $n = 7736$) showed enrichment of neuronal and synaptic proteins confirming neuron-specific labeling. **h** Graphical representation of the number of proteins within various cellular compartments determined from GSEA. For related MS data and additional analyses, see Supplementary Data 1, 3, 7, and 8. Source data are provided as a Source Data file.

Supplementary Data 2), highlighting the ability of CIBOP to capture abundant phosphorylated proteoforms even without phosphopeptide enrichment.

Gene set enrichment analyses (GSEA) of the 2143 enriched biotinylated proteins revealed enrichment of neuronal and synaptic proteins involved in neuron projection and synapse organization, confirming neuron-specificity of labeling (Fig. 1g, h, Supplementary Data 3). GSEA also showed labeling of proteins involved in a diverse set of molecular functions such as cytoskeletal protein binding, substrate-specific transporter activity, and calmodulin binding. Specifically, we observed biotinylation of 54 cell surface proteins, 63 transmembrane transporters, and 45 ion channel subunits (7 calcium, 9 glutamate, 6 GABA, 13 potassium, 1 sodium, 4 anionic) (Supplementary Data 4). The biotinylated proteome was enriched in cytoplasmic, membrane, synaptic, and cytoskeletal proteins, which are representative of the whole cell proteome rather than a bias to a specific subcellular compartment (Fig. 1h, Supplementary Data 3). In comparison to bulk brain proteomic data from the AAV cohort, our pan-neuronal enriched proteome identified 354 neuron-derived proteins (Supplementary Data 1) that were not previously quantified at the whole brain level. GSEA of these 354 neuron-derived proteins showed that they were predominantly membrane proteins involved in a variety of molecular functions such as serotonin receptor, potassium channel, glutamate receptor, and protein kinase activity (Supplementary Data 6). Within the 354 proteins, we identified 33 druggable targets of disease relevance (Supplementary Data 1), including solute transporters, GPCRs (e.g., GRM1), lipid metabolic proteins (e.g., GBA2), and signaling proteins including AKT1 which is a key member of the IGF1/PI3K/AKT1/mTOR axis that is relevant to synaptic functioning and memory[31]. We also performed all pairwise comparisons between labeled and control mice and obtained nearly identical GSEA results as above (Supplementary Fig. 1d, e and Supplementary Data 1, 7, 8).

In sum, AAV-CIBOP resulted in robust pan-neuronal labeling of proteins in the hippocampus by TurboID in vivo. Proteomic analysis of biotinylated proteins confirmed neuronal enrichment and representation of proteins within a diverse number of molecular functions from various cellular compartments, including numerous synaptic proteins, transmembrane proteins, and several druggable targets which are otherwise challenging to sample in the native state of neurons in adult mouse brain.

**Camk2a-neuronal protein biotinylation in adult mouse brain using a transgenic strategy.** We next employed a transgenic approach to express TurboID within Camk2a neurons by breeding

*Rosa26*^TurboID/wt^ mice with Camk2a-Cre^Ert2^ and inducing Cre recombinase expression by intraperitoneal tamoxifen. Camk2a (Ca^2+^/calmodulin-activated protein kinase 2A) is an abundant serine-threonine kinase highly expressed by excitatory neurons, particularly in the synapse, where it regulates synaptic transmission, excitability, and long-term potentiation. Camk2a was chosen based on extensive validation, specificity, and non-leakiness of available Camk2a-Cre^Ert2^ driver lines and the well-characterized expression patterns of Camk2a across brain regions[26,27]. *Rosa26*^TurboID/wt^/Camk2a and littermate controls received tamoxifen at 6 weeks of age, allowed 3 weeks for recombination, and followed by biotin supplementation for 2 weeks (Fig. 2a). There were no associated phenotypic changes or observable adverse effects during biotin supplementation (e.g., weight and locomotor activity). Western blot analysis of lysates from different brain regions (cortex, hippocampus, striatum/thalamus, pons/medulla, and cerebellum) confirmed robust biotinylation of proteins in the *Rosa26*^TurboID/wt^/Camk2a mice as compared to minimal endogenous biotinylation observed in control mice (Fig. 2b). Qualitatively, the highest level of labeling was observed in the cortex, hippocampus, and striatum/thalamus regions as compared to cerebellum and pons/medulla, consistent with known Camk2a expression patterns[32]. We also confirmed TurboID protein expression via detection of V5 in *Rosa26*^TurboID/wt^/Camk2a brain regions only, which followed a similar pattern to level of biotinylation (Fig. 2b).

Immunofluorescent imaging of the whole brain displayed wide-spread biotinylation in *Rosa26*^TurboID/wt^/Camk2a brains compared to control brains (Fig. 2c). Furthermore, biotinylation was observed predominantly within cell bodies and axonal projections in all brain regions sampled for proteomic analysis (Fig. 2c). Map2 and streptavidin co-immunofluorescence confirmed neuronal labeling throughout the hippocampus (Fig. 2d), as well as the cortex (Supplementary Fig. 2a), pons/medulla (Supplementary Fig. 2b), striatum/thalamus (Supplementary Fig. 2c), and cerebellum (Supplementary Fig. 2d). Importantly, biotinylation was not observed in astrocytes (Supplementary Fig. 3a) or microglia (Supplementary Fig. 3b), and there was no evidence of reactive gliosis in *Rosa26*^TurboID/wt^/Camk2a mice compared to control mice (Supplementary Fig. 3).

To determine whether TurboID expression and protein biotinylation impact Camk2a neuronal function, we performed electrophysiological studies on 3-month-old *Rosa26*^TurboID/wt^/Camk2a mice and littermate controls that received tamoxifen and biotin supplementation (Fig. 2e). Somatic whole-cell recordings from pyramidal neurons in the CA3c region of the hippocampus,

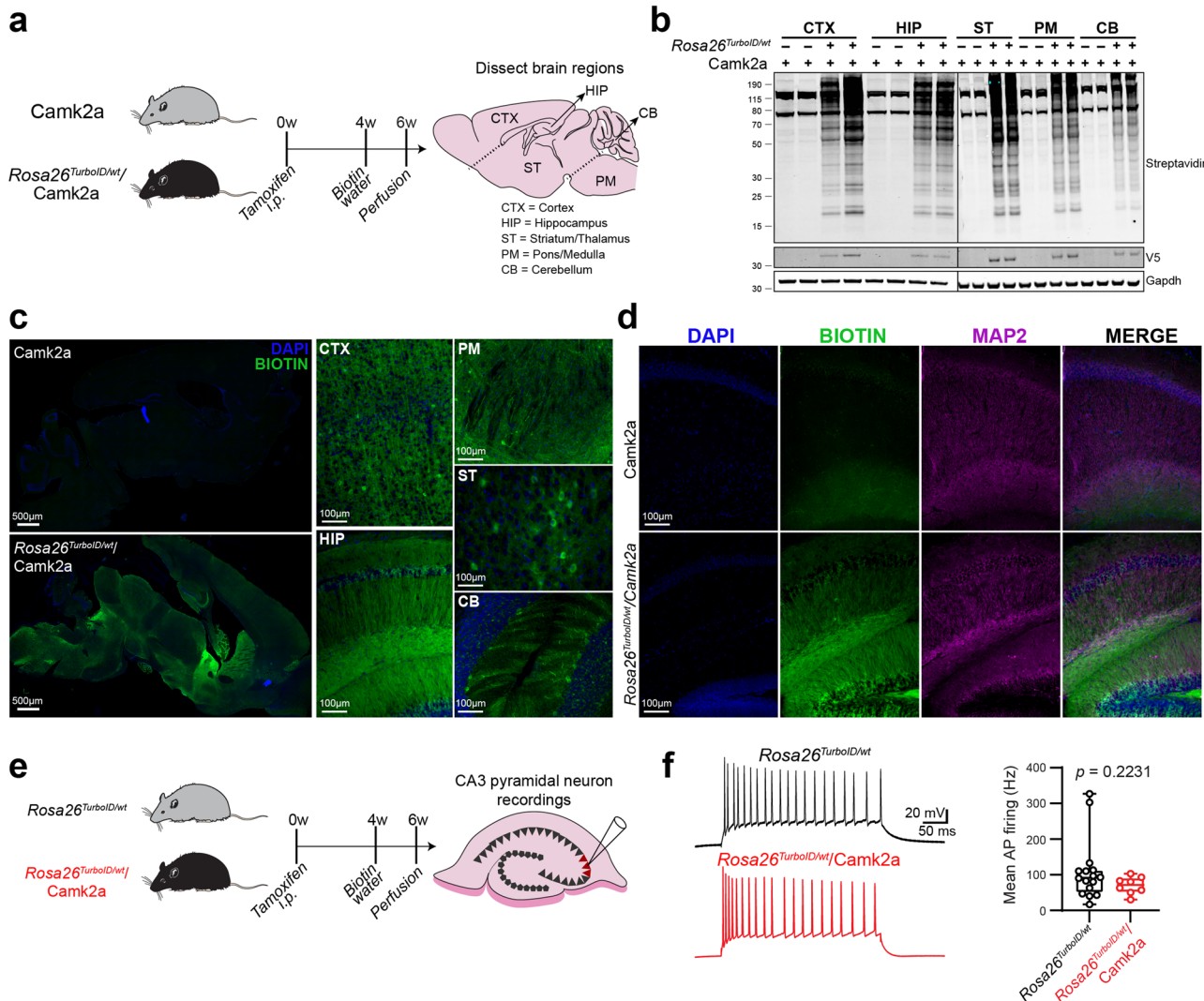

**Fig. 2 Successful biotin labeling of Camk2a neurons in adult mouse brain. a** Schematic displaying the study design for Camk2a neuron-specific proteomic biotinylation. Heterozygous Camk2a-Cre[Ert2] ($n = 2$) and $Rosa26^{TurboID/wt}$/Camk2a-Cre[Ert2] mice ($n = 2$) received tamoxifen intraperitoneally (i.p.) once a day for 5 days. After waiting 4 weeks for sufficient recombination and washout of tamoxifen effect, mice were supplemented with biotin water (37.5 mg/L) for 2 weeks. Brain was dissected into the following regions: cortex (CTX), hippocampus (HIP), striatal region which included the thalamus (ST), brain stem comprised on pons/medulla (PM), and cerebellum (CB). **b** Representative Western blots from brain lysates ($n = 2$ mice/experimental group), probed for biotin (streptavidin fluorophore), V5 (to detect V5-TurboID-NES), and Gapdh are shown. $Rosa26^{TurboID/wt}$/Camk2a brain regions showed biotinylated proteins of different molecular weights as compared to few endogenously biotinylated proteins in the control brain regions. **c** Representative immunofluorescence images ($n = 2$ mice/experimental group) showing biotinylation (green: streptavidin Alexa488) across different brain regions. Tiled images of the entire hemisphere (sagittal section) from control and labeled mice, and higher-magnification images from individual regions of labeled mice are shown. Nuclei were labeled with DAPI (blue). **d** Representative immunofluorescence images ($n = 2$ mice/experimental group) showing overlap between biotinylation (streptavidin Alexa488) and Map2 expression in neurons and axons in the hippocampus from control and labeled mice. **e** Experimental outline for hippocampal slice electrophysiology from non-labeled $Rosa26^{TurboID/wt}$ and labeled $Rosa26^{TurboID/wt}$/Camk2a mice ($n = 2$ mice/group). **f** Electrophysiological recordings from CA3c pyramidal neurons of the hippocampus showing similar firing pattern in controls and labeled $Rosa26^{TurboID/wt}$/Camk2a mice. Each data point represents a single neuron. Pooled analysis from $n = 17$ non-labeled control and $n = 8$ labeled neurons is shown on the right ($p$ value represents unpaired $t$-test, $p = 0.2231$). Data are represented as box plots, indicating median, inter-quartile range, 10th and 90th percentile. For associated immunofluorescence images for confirmation of efficient biotin labeling and additional electrophysiological studies performed, see Supplementary Fig. 4. The figure was partly generated using Servier Medical Art, provided by Servier, licensed under a Creative Commons Attribution 3.0 unported license. Source data are provided as a Source Data file.

a region that displayed robust biotinylation (Supplementary Fig. 4a), showed no significant differences in mean action potential firing between control and $Rosa26^{TurboID/wt}$/Camk2a mice (Fig. 2f). Furthermore, we did not observe significant differences in various passive and active parameters measured (Supplementary Fig. 4b). In summary, these findings validate the Tg-CIBOP approach and confirm lack of any electrophysiological perturbations in Camk2a neurons despite robust

proteomic biotinylation as well as the lack of phenotypic changes in the mouse.

**Proteomic analysis reveals unique Camk2a neuron brain region signatures**. After confirming biotinylation of proteins by Western blot and immunofluorescence imaging in $Rosa26^{TurboID/wt}$/Camk2a brains, we enriched biotinylated proteins from cortex, hippocampus,

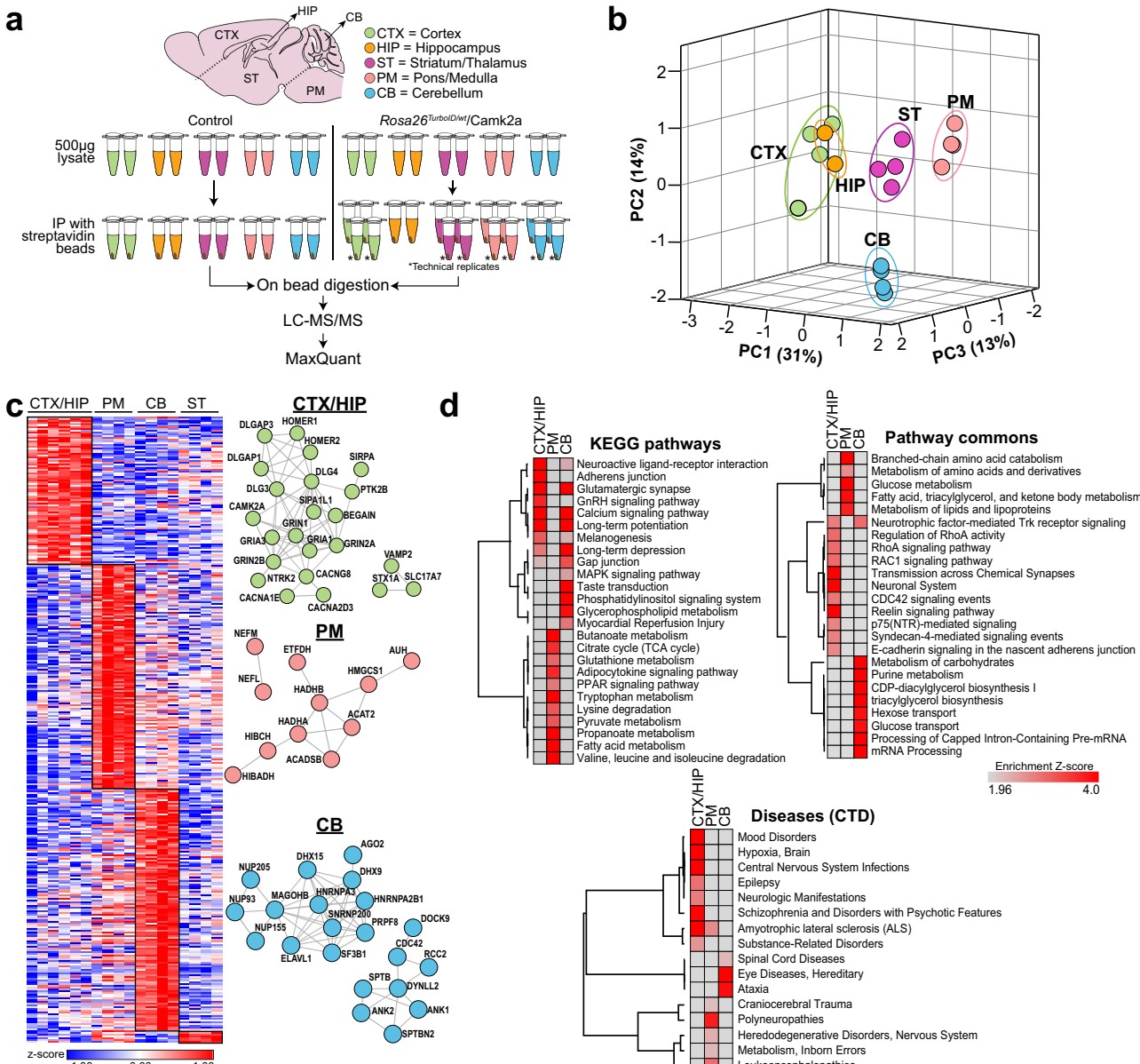

**Fig. 3 Camk2a neurons exhibit region-specific proteomic differences in adult mouse brain. a** Experimental outline for LFQ-MS studies performed on biotinylated proteins enriched using streptavidin beads from *Rosa26^TurboID/wt^*/Camk2a-Cre^Ert2^ and littermate control (Camk2a-Cre^Ert2^) mice (*n* = 2 mice per experimental group, five regions per mouse with 2 technical replicates per region). Asterisk (*) indicates technical replicates for each brain region from *Rosa26^TurboID/wt^*/Camk2a-Cre^Ert2^ mice. **b** Principal Component Analysis (PCA) of MS data after normalization to TurboID abundance in each sample/brain region. PCA identified distinct clusters based on region except for hippocampal and cortical regions clustering together. Three PCs explained 30%, 14 and 13% of variance, respectively. **c** Clustering representation of protein abundance data of core groups of proteins most highly expressed in specific brain regions with at least 4-fold higher levels in a specific region compared to all other regions (*p* ≤ 0.05). STRING analysis identified networks of known direct (protein-protein) and indirect (functional) interactions within core regional protein signatures. **d** Heatmap representation, based on enrichment Z-scores, of KEGG pathways, Pathway Commons, and diseases from the Comparative Toxicogenomics Database (CTD) enriched in core regional proteins. For related MS data and additional analyses, see Supplementary Data 9, 11, 12, and 13. The figure was partly generated using Servier Medical Art, provided by Servier, licensed under a Creative Commons Attribution 3.0 unported license.

striatum/thalamus, pons/medulla, and cerebellum using streptavidin beads, followed by LFQ-MS (Fig. 3a). We quantified 2096 unique proteins across all samples. Differential expression analysis of proteomic data from all brain regions, after normalizing for TurboID levels across brain regions, comparing *Rosa26^TurboID/wt^*/Camk2a and controls resulted in 1269 proteins with ≥ 2-fold enrichment while only four proteins showed ≥ 2-fold enrichment in the control brains (Supplementary Fig. 5a, Supplementary Data 9). GSEA of 1245 enriched proteins identified those involved in biological

processes such as transport, cytoskeletal organization, and cellular membrane organization as well as proteins involved in molecular functions such as cytoskeletal protein binding, glutamate receptor activity, and clathrin binding (Supplementary Fig. 5b, Supplementary Data 10). Proteins within the cytoplasm, membrane, neuronal cell body, synapse, and other cellular components were significantly enriched in the *Rosa26^TurboID/wt^*/Camk2a proteome (Supplementary Fig. 5c). In comparison to a reference bulk brain proteomic dataset from adult mouse brain[5], our Camk2a neuronal proteome identified

114 neuron-derived proteins that were not previously quantified at the whole brain level (Supplementary Data 9).

Principal component analysis (PCA) of $Rosa26^{TurboID/wt}$/Camk2a biotinylated protein abundance data showed that nearly 60% of variance was explained by 3 principal components (PCs), of which PC1 explained 30% while PC2 explained 14% and PC3 explained 13% of variance (Fig. 3b). Notably, PCA identified four distinct brain regional proteomic clusters: cortex and hippocampus, striatum/thalamus, pons/medulla, and cerebellum (Fig. 3b). We identified core groups of highly expressed region-specific proteomic signatures of Camk2a neurons ($\geq$ 4-fold change and $p \leq 0.05$, Fig. 3c, Supplementary Data 11). The cortex/hippocampus contained 83 signature proteins while cerebellum contained 138, pons/medulla contained 128, and striatum/thalamus contained 6. We also identified networks of known direct (protein-protein) and indirect (functional) interactions (STRING) within these core regional protein signatures (Fig. 3c).

GSEA of core protein signatures revealed distinct molecular and biological features of Camk2a neurons (Supplementary Data 12). The proteomic signature of cortical and hippocampal neurons was indicative of glutamatergic synaptic transmission, calcium signaling, synaptic plasticity, and more specifically, E-cadherin and syndecan-4-mediated signaling and RhoA activity (Fig. 3d). Cortical and hippocampal neuronal signature proteins were enriched in genes associated with epilepsy, mood and psychotic disorders, substance abuse, and hypoxic brain injury (Fig. 3d). The cerebellar Camk2a proteomic signature was indicative of increased MAPK signaling and mRNA processing (splicing, snRNP assembly) as well as genes related to ataxia, spinal cord disease, and hereditary eye diseases (e.g., retinitis pigmentosa) (Fig. 3d). Unlike cortical and cerebellar neurons, Camk2a neurons from the pons/medulla showed increased levels of proteins involved in amino acid, fatty acid, and carbohydrate metabolism (Fig. 3d), supportive of elevated metabolic activity in these neurons and axons. Consistent with the abundant metabolic protein signature of pons/medullary Camk2a neurons and the axonal predominance in this region, this regional proteomic signature was enriched in gene symbols related to inborn errors of metabolism, leukoencephalopathies, and polyneuropathies (Fig. 3d). Since ion channels determine the electrophysiological properties of neurons, we also identified distinct ion channel subunits that showed regional specificity. In the cortex and hippocampus, core ion channel proteins included voltage gated calcium channel subunits Cacna1e, Cacna2d3, and NMDA receptor subunit Grin1. Cerebellar Camk2a neuronal signatures were characterized by higher levels of voltage-gated calcium channel subunits Cacna1a, Cacna2d2, Cacng2, chloride channel Clic1, and several potassium channels including Kcnc1, Kcnc3, Kcnd2, Kcnip1. In contrast, pons/medulla Camk2a neurons were characterized by higher expression of potassium channels Hcn2, Kcna2 and Kcnj10. Similar to the AAV-CIBOP results, we identified several ($n = 52$) druggable target proteins that exhibited region-specific patterns (Supplementary Data 11). Drug targets with high levels in cortical Camk2a neurons included ion channel Gria3, a regulator of neurogenesis (Ntrk2), and lipid metabolic protein Plcb1. In contrast, drug targets in the cerebellum included mRNA transport and mRNA processing proteins (e.g., Snrnp200) and the phospholipase Plcb4.

Region specific proteins were further searched against curated lists of gene-disease associations (DisGeNet) to define regional gene-disease associations (Supplementary Data 13). Cortex and hippocampal Camk2a neuron proteins enriched in glutamatergic transmission proteins and long-term potentiation (e.g., Lingo1, Homer1, Gria1) were associated with mental health disorders (schizophrenia, bipolar disease, depression, autism), neurodevelopmental disorders, essential tremor, and primary epileptic disorders (e.g., West

syndrome). Pons/medulla enriched proteins were associated with dystonia, Parkinson's disease, polyneuropathies, and inborn errors in metabolism consistent with the high metabolic protein levels in this region. Cerebellar Camk2a neuronal proteins enriched in mRNA processing and splicing proteins were linked to ataxias, primary lateral sclerosis, and psychiatric disorders, suggesting that splicing dysregulation in cerebellar Purkinje neurons may underlie pathogenesis of these conditions.

Since the above analyses were targeted towards proteins with region-specific enrichment patterns, we also analyzed our data in an unbiased manner (K-means clustering). As a result, we identified five clusters of proteins that showed regional patterns of protein expression (Supplementary Fig. 6a, Supplementary Data 14) consistent with region-specific analyses described above. GSEA of the clusters revealed that cortical and hippocampal Camk2a neurons expressed higher levels of proteins involved in GABA signaling and glutamate signaling pathways, neuron development, and synaptic function (Cluster 2 & 4, Supplementary Fig. 6b, Supplementary Data 15) while the cerebellar Camk2a neuronal proteome showed higher levels of proteins involved in hydrolase activity and translation initiation factor 3 (Cluster 5, Supplementary Fig. 6b, Supplementary Data 15). The pons/medulla Camk2a proteome showed a unique signature of pigment granule as well as tau-kinase activity related proteins. GSEA also indicated regional metabolic differences in Camk2a neurons, with over-representation of glycolytic, amino acid, and fatty acid metabolic proteins in the pons/medulla (Cluster 1 & 3, Supplementary Fig. 6b, Supplementary Data 15).

We also quantified the whole brain regional proteomes (background) from $Rosa26^{TurboID/wt}$/Camk2a and control mice. A total of 3969 proteins were quantified and displayed regional enrichment (Supplementary Fig. 7a, Supplementary Data 16). Among the 3969 proteins, 1872 were also identified in the Camk2a-enriched proteome with 201 proteins being exclusive to the Camk2a-enriched proteome (Supplementary Fig. 7b). Using definitions of core regional protein markers ($\geq$4-fold enrichment in the specific region over other regions and $p \leq 0.05$), 1110 proteins were identified as core-markers in the Camk2a-enriched proteome, of which only 343 were also identified as core markers in the background proteome (Supplementary Fig. 7c). When we assessed the degree of overlap between each region, we found that the Camk2a-enriched proteome contained 549 regionally enriched proteins in the cortex/hippocampus, of which 419 (76.3%) were also regionally enriched in the background proteome (Supplementary Fig. 7d). In contrast, lower levels of overlap were observed in other regions - 9.6% in the striatum/thalamus, 58% in the pons/medulla, and 59% in the cerebellum (Supplementary Fig. 7e–g). This shows that the Camk2a-enriched proteome identifies twice as many core-regional proteomic differences within Camk2a neurons that were not captured by the background proteome.

In sum, Tg-CIBOP allowed us to comprehensively characterize the proteome of Camk2a neurons in the adult mouse brain and resolve regional differences in proteomic composition of Camk2a neurons, which was not easily attainable with the AAV approach or from mouse post-natal or embryonic neuronal culture systems. Importantly, we uncovered potential links between regional proteomic characteristics of Camk2a neurons and disease vulnerability.

**Regional Camk2a neuron-derived phospho-protein signaling and cytokine signatures in adult mouse brain.** MS based quantitative proteomics provide a comprehensive and unbiased molecular snapshot of the cell; although, these approaches often fail to detect small and less-abundant proteins such as cytokines

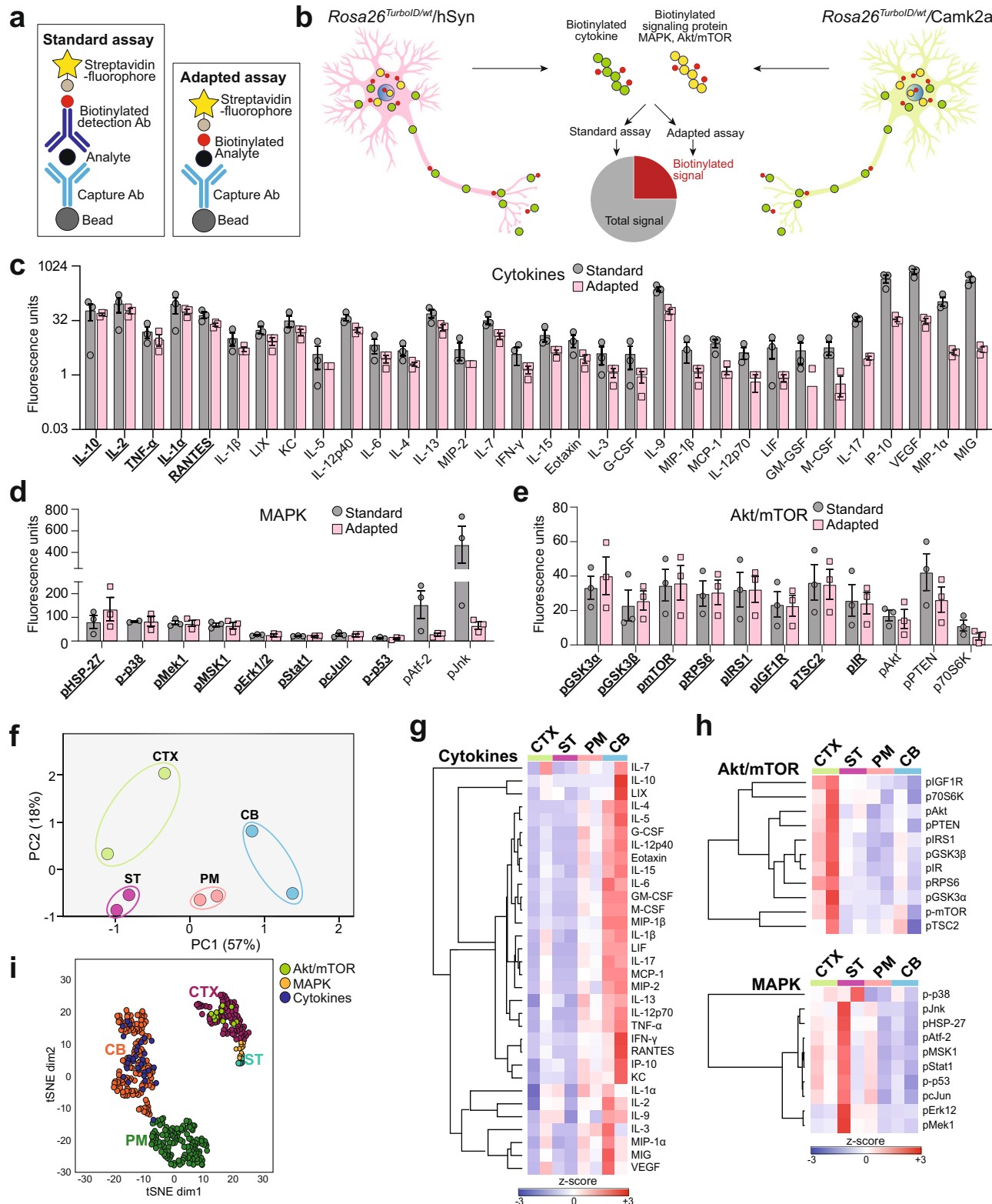

and signaling proteins without prior enrichment. Thus, we extended our studies to measure specific biotinylated cytokines and phospho-proteins involved in cellular signaling cascades (e.g., MAPK and Akt/mTOR) using an antibody-based approach. We adapted the multiplexed Luminex sandwich ELISA approach (Fig. 4a) to directly measure biotinylated phospho-proteins from the MAPK and Akt/mTOR pathways and an array of 32 inflammatory cytokines in brain tissues from both $Rosa26^{TurboID/wt}$ AAV and

transgenic cohorts (Fig. 4b). First, we captured the proteins of interest onto beads using target-specific antibodies, then a second biotinylated detection antibody followed by a streptavidin fluorophore and fluorescence quantitation (Fig. 4a, standard assay). In parallel, we adapted this assay to exclude the second biotinylated detection antibody to directly detect neuronal biotinylated proteins of interest via streptavidin fluorophore (Fig. 4a, adapted assay). This provided an estimate of total (standard assay) and neuron-derived

**Fig. 4 Neuron-specific regional differences in cellular signaling and cytokine levels. a** Cartoon showing standard and adapted Luminex methods to measure total and biotinylated proteins. **b** Cartoon representation of standard and adapted Luminex approaches to measure cytokines, MAPK and Akt/mTOR phospho-proteins in brain lysates from $Rosa26^{TurboID/wt}$ AAV and transgenic cohorts. **c–e** Fluorescence values for cytokines, MAPK and Akt/mTOR phospho-proteins measured using standard and adapted assays, from $Rosa26^{TurboID/wt}$/AAV9-hSyn-Cre mice ($n = 3$/group). Predominantly neuron-derived cytokines or phospho-proteins have similar fluorescence values in both standard and adapted assays, while predominantly non-neuronal cytokines or phospho-proteins have a markedly lower adapted assay readout as compared to the standard assay, determined by pairwise comparison between control and labeled mice (also see Supplementary Data 17). Data shown as mean ± SEM. **f** PCA of adapted Luminex assays performed on brain lysates from the $Rosa26^{TurboID/wt}$ transgenic cohort, showing regional differences (CTX, ST, PM, and CB) based on cytokines and MAPK and Akt/mTOR phospho-proteins. Fluorescence values from adapted assays from labeled ($Rosa26^{TurboID/wt}$/Camk2a-Cre$^{Ert2}$) and non-labeled controls (Camk2a-Cre$^{Ert2}$) brain lysates were normalized to TurboID abundance from MS studies after which signal from adapted assays from control mice was subtracted. **g, h** Heat map summarizing HCA of normalized data from adapted Luminex assays. Analyses of cytokines and phospho-proteins from MAPK and Akt/mTOR pathways are shown, revealing region specific signatures. Overall, neuron-derived cytokines had higher levels in the cerebellum and relatively lower levels in the cortex while phospho-proteins, particularly from the Akt/mTOR pathway had higher levels in the cortex and lower levels in the cerebellum. Related raw and normalized Luminex data from all animals from AAV and transgenic cohorts are included in Supplementary Data 19–21. **i** An integrated tSNE of core regional proteomic signatures and Luminex data showed clustering of Akt/mTOR signaling with cortex-specific proteins while elevated cytokines clustered with the cerebellar proteomic signature and MAPK clustered with striatal/thalamic proteomic signature. (see Supplementary Data 18 for statistics). The figure was partly generated using Servier Medical Art, provided by Servier, licensed under a Creative Commons Attribution 3.0 unported license. Source data are provided as a Source Data file.

(adapted assay) levels of the target protein without the need for enrichment of biotinylated proteins.

In the AAV cohort, the standard Luminex assay showed that AAV9-hSyn-Cre increased levels of signaling phospho-proteins (pAkt, pPTEN) and inflammatory cytokines (IL-2, IL-13, IL-17, IP-10, KC, and MIG) indicating an inflammatory response related to AAV9 injection (Supplementary Data 17). Given this effect of AAV9, we subsequently compared $Rosa26^{TurboID/wt}$/hSyn mice to WT/hSyn mice using the standard assay and found significant elevation of 10 phospho-proteins (e.g., pMSK1, pStat1, pMek1) and 9 cytokines (e.g., VEGF, MIG, IL7) in labeled mice. This increased phospho-protein and cytokine response in $Rosa26^{TurboID/wt}$/hSyn mice could be due to either an inflammatory response or due to additive increased signal of biotinylated target proteins in the assay. Despite these differences between groups using the standard assay, the adapted Luminex approach detected 9-fold higher signal in the $Rosa26^{TurboID/wt}$/hSyn brains compared to minimal signal in the control brains, confirming that the elevated signal was indeed due to biotinylation by TurboID. By comparing the standard assay signal (total analyte abundance) with the adapted assay signal (biotinylated neuron-derived signal) in the $Rosa26^{TurboID/wt}$/hSyn brain lysates, we found that majority of cytokines, with the exceptions of IL-10, IL-2, TNF-α, IL-1α, and RANTES (Fig. 4c, underlined proteins), had adapted assay signals less than 50% of the standard assay signals, suggesting their predominant origin from non-neuronal cells. We also quantified several MAPK phosphoproteins (Fig. 4d, underlined proteins, e.g., p-p38, pMek1, and pErk1/2) and Akt/mTOR phosphoproteins (Fig. 4e, underlined proteins, e.g., pGSK-α and -β, p-mTOR), and observed that >50% of the standard assay signal for these analytes, was accounted for by the adapted signal, indicating their predominant origin from neurons.

We also applied the adapted Luminex approach to investigate regional differences in neuron-derived signaling proteins and cytokines in $Rosa26^{TurboID/wt}$/Camk2a mice. After accounting for background, adapted Luminex assay signals were negligible in the control mice and were subtracted (Supplementary Data 18). Highest signal intensities in the adapted assay across all regions were observed for 25 of 32 cytokines (Top 5: IL-2, IL-1α, IL-9, IP-10, and VEGF) and phospho-proteins from both Akt/mTOR (11 of 11 analytes) and MAPK (10 of 10) signaling pathways. Similar to the results from our AAV cohort, phosphoproteins from signaling pathways were more robustly biotinylated as compared to cytokines (Supplementary Data 18). A combined PCA of fluorescent intensities from adapted Luminex assays for MAPK,

Akt/mTOR, and cytokine panels revealed four distinct clusters associated with brain regions of $Rosa26^{TurboID/wt}$/Camk2a mice (Fig. 4f). Specifically, we identified unique patterns of neuron-derived cytokine levels that distinguished the cerebellum from the cortex and pons/medulla after adjusting for TurboID levels (Fig. 4g). Intriguingly, cortical Camk2a neuron-derived signature included low levels of cytokines but high levels of phospho-proteins from the Akt/mTOR pathway (Fig. 4h). In contrast, cerebellar Camk2a neuronal signature displayed higher level of cytokines and low levels of Akt/mTOR and MAPK phospho-proteins, while MAPK phospho-proteins appear to be enriched in the striatum/thalamus. (Fig. 4g, h). An integrated t-distributed stochastic neighbor embedding (tSNE) analysis of core regional proteomic signatures derived from LFQ-MS studies and from Luminex data confirmed clustering of Akt/mTOR signaling with cortex-specific proteins while elevated cytokines clustered with the cerebellar proteomic signature (Fig. 4i). To ensure that signals detected by the adapted assay are only due to protein biotinylation, we performed pre-blocking studies using monomeric avidin before addition of the detection antibody, which completely abolished the adapted signal from $Rosa26^{TurboID/wt}$/Camk2a brains (Supplementary Fig. 8).

Collectively, these targeted immunoassays of biotinylated proteins provide a complementary approach to MS for detecting proteomic signatures that resolve signaling mechanisms underlying cell-type specific phenotypes in brain.

**Astrocyte protein biotinylation reveals region-specific proteomic signatures and differences between Camk2a neurons in adult mouse brain.** Based on our successful application of Tg-CIBOP to Camk2a neurons, we extended this approach to Aldh1l1 astrocytes and contrasted the proteomic profiles of the two cell types in their native state. To achieve astrocyte-specific proteomic labeling, we bred $Rosa26^{TurboID/wt}$ mice with Aldh1l1-Cre$^{Ert2}$ mice[28,29], a well-validated inducible Cre mouse line, to target mature astrocytes in the brain and applied the same tamoxifen and biotinylation paradigms as above (Fig. 5a). $Rosa26^{TurboID/wt}$/Aldh1l1, $Rosa26^{TurboID/wt}$/Camk2a (independent of those used in Figs. 2–3), and littermate $Rosa26^{TurboID/wt}$ control mice were used for experiments. After 2 weeks of biotinylation, brain regions (cortex, hippocampus, striatum/thalamus, pons/medulla, cerebellum, and spinal cord) were dissected from one hemisphere for proteomic studies, while the other hemisphere was used for immunohistochemical studies. Western blot analysis of lysates from different brain regions of $Rosa26^{TurboID/wt}$/Aldh1l1 and

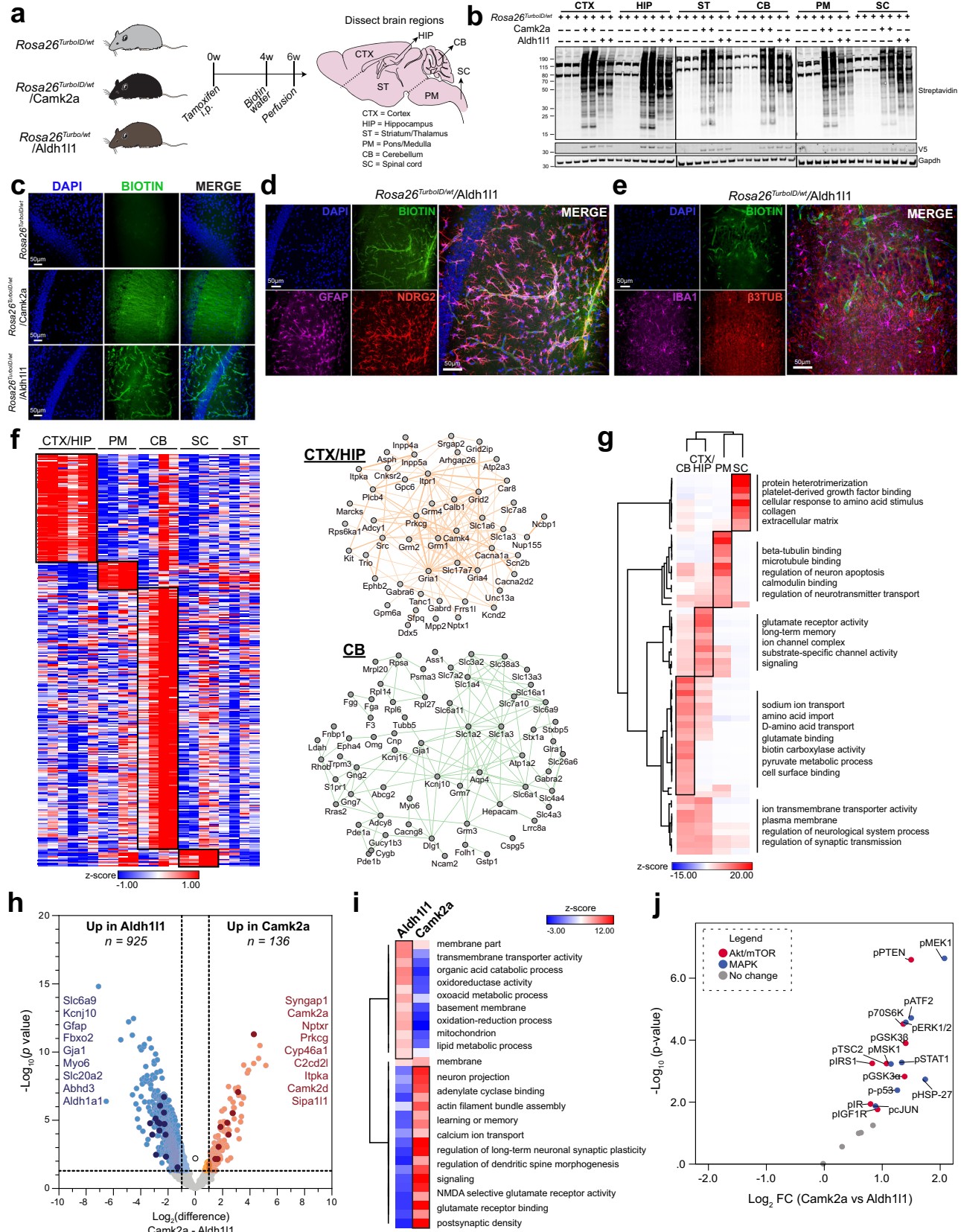

*Rosa26^{TurboID/wt}*/Camk2a mice confirmed biotinylation of proteins with distinct patterns of biotinylation between the two mouse lines, with few endogenously biotinylated proteins observed in control mice (Fig. 5b). Qualitatively, *Rosa26^{TurboID/wt}*/

Camk2a brain regions displayed a higher degree of labeling compared to *Rosa26^{TurboID/wt}*/Aldh1l1 brain regions. We also confirmed TurboID protein expression via detection of V5 (Fig. 5b). Immunofluorescent imaging of brain sections from

**Fig. 5 Astrocyte protein biotinylation reveals region-specific proteomic signatures and differences between Camk2a neurons. a** Study design for neuron-specific and astrocyte-specific proteomic biotinylation. $Rosa26^{TurboID/wt}$, $Rosa26^{TurboID/wt}$/Camk2a-Cre$^{Ert2}$, $Rosa26^{TurboID/wt}$/Aldh1l1-Cre$^{Ert2}$ mice received tamoxifen intraperitoneally for 5 days. After 4 weeks, mice received biotin water for 2 weeks. **b** Representative Western blots from brain region lysates ($n = 2$ mice/group), probed for biotin (streptavidin Alexa488), V5 and Gapdh are shown. **c** Representative images ($n = 2$ mice/group) showing biotinylation in the hippocampus. **d** Representative immunofluorescence images ($n = 2$ mice/group) showing overlap between astrocytic biotinylation (streptavidin Alexa488), Gfap, and Ndrg2 in the hippocampus region from $Rosa26^{TurboID/wt}$/Aldh1l1 mice. **e** Representative immunofluorescence images ($n = 2$ mice/group) showing no overlap between biotinylation, Iba1, and βIII-tubulin in astrocytes and blood vessels in the hippocampus region from $Rosa26^{TurboID/wt}$/Aldh1l1 mice. **f** Clustering analysis of protein abundance data of region-enriched proteins with at least 4-fold enrichment over other regions in $Rosa26^{TurboID/wt}$/Aldh1l1 mice. STRING analysis identified networks of direct (protein-protein) and indirect (functional) interactions within core regional protein signatures in cortex/hippocampus and cerebellum. **g** Heatmap representation, based on enrichment Z-scores, of gene ontologies enriched in core regional proteins. **h** Volcano plot showing differentially enriched proteins comparing $Rosa26^{TurboID/wt}$/Aldh1l1 and $Rosa26^{TurboID/wt}$/Camk2a mice. For this analysis, all six brain regions were combined for both groups. Orange symbols (two-tailed $T$ test unadjusted $p \leq 0.05$ and $\geq$ 2-fold change) represent biotinylated proteins enriched in Camk2a neurons with neuron-specific proteins highlighted. Blue symbols (two-tailed $T$ test unadjusted $p \leq 0.05$ and $\geq$ 2-fold change) represent biotinylated proteins enriched in Aldh1l1 astrocytes and examples of astrocyte-specific proteins are highlighted in dark blue. **i** HCA of GSEA showing over-represented ontologies within neuronal-enriched proteins and astrocyte-enriched proteins. Representative gene ontology terms are highlighted. **j** Volcano plot showing enrichment (two-tailed $T$ test unadjusted $p \leq 0.05$) of MAPK and Akt/mTOR phospho-proteins in $Rosa26^{TurboID/wt}$/Camk2a compared to $Rosa26^{TurboID/wt}$/Aldh1l1 brains. For related MS data and additional analyses, see Supplementary Data 19–24. The figure was partly generated using Servier Medical Art, provided by Servier, licensed under a Creative Commons Attribution 3.0 unported license. Source data are provided as a Source Data file. Source data are provided as a Source Data file.

these mice demonstrated biotinylation, as detected by Streptavidin-488, in $Rosa26^{TurboID/wt}$/Camk2a and $Rosa26^{TurboID/wt}$/Aldh1l1 brains while control brains lacked any biotinylation (Fig. 5c). Biotinylation in $Rosa26^{TurboID/wt}$/Aldh1l1 brains colocalized with Gfap- and Ndrg2- positive astrocytes with a preponderance of labeled blood vessels in the hippocampus (Fig. 5e) as well as other brain regions (Supplementary Fig. 9), with no evidence of gliosis (Fig. 5e). Furthermore, biotinylation was not observed in βIII-tubulin-positive neurons or Iba1-positive microglia, confirming astrocyte specificity in the $Rosa26^{TurboID/wt}$/Aldh1l1 brains (Fig. 5d). Given the prevalence of biotin labeling around and within blood vessels, we hypothesized that this signal may be due to astrocytic end-feet that are in close contact with blood vessels[33]. To confirm this, we immunolabeled the same brain sections for aquaporin 4 (AQP4), an aquaporin channel specifically located at the perivascular astrocyte foot processes, GFAP, and biotin. We observed global immunopositivity for AQP4 and GFAP in control and $Rosa26^{TurboID/wt}$/Aldh1l1 hippocampi, with biotin signal observed only in the $Rosa26^{TurboID/wt}$/Aldh1l1 brains (Supplementary Fig. 10a). At a higher magnification, we saw colocalization of biotin with AQP4 as well as GFAP (Supplementary Fig. 10b). Lastly, biotinylation in $Rosa26^{TurboID/wt}$/Camk2a mice were similar to prior findings discussed above (Supplementary Fig. 9f).

We enriched biotinylated proteins from $Rosa26^{TurboID/wt}$/Camk2a and $Rosa26^{TurboID/wt}$/Aldh1l1 brain regions and performed LFQ-MS, quantifying 3216 unique proteins. After enforcing a missing value threshold and imputing missing values, 2550 proteins remained in our analyses. First, we performed differential expression analysis, after normalizing for TurboID levels, from all brain regions comparing $Rosa26^{TurboID/wt}$/Aldh1l1 and controls. As a result, 1380 proteins with $\geq$ 2-fold enrichment in $Rosa26^{TurboID/wt}$/Aldh1l1 brains and 21 proteins with $\geq$ 2-fold enrichment in the control brains were identified (Supplementary Fig. 11a, Supplementary Data 19). GSEA of 1380 enriched proteins identified those involved in biological processes such as membrane organization and metabolic activity as well as proteins involved in molecular functions such as transporter activity, mitochondrial function, and calbindin binding (Supplementary Fig. 11b, Supplementary Data 20). Proteins within the cytoplasm, membrane, and other cellular components were significantly enriched in the $Rosa26^{TurboID/wt}$/Aldh1l1 proteome (Supplementary Fig. 10b, Supplementary Data 20).

Next, we analyzed the $Rosa26^{TurboID/wt}$/Aldh1l1 proteome for regional differences, complementary to the region-resolved neuronal proteomics described above and identified core groups of astrocyte-derived proteins that were highly abundant in a region-specific manner ($\geq$ 4-fold change and $p \leq 0.05$, Fig. 5f, Supplementary Data 21). We also identified networks of known direct (protein-protein) and indirect (functional) interactions (STRING) within core cortex/hippocampus and cerebellum protein signatures (Fig. 5f). GSEA of these core regional astrocyte proteins revealed enrichment of metabolic processes in the cerebellum, glutamate receptor and ion channel activity in the cortex/hippocampus, microtubule and calmodulin binding in the pons/medulla, and protein heterodimerization and extracellular matrix in the spinal cord (Fig. 5g, Supplementary Data 22).

We next compared the proteomes from $Rosa26^{TurboID/wt}$/Camk2a and $Rosa26^{TurboID/wt}$/Aldh1l1 brain regions and identified 1061 proteins with cell type-specific differences ($\geq$ 2-fold change and $p \leq 0.05$, Fig. 5h), including 925 proteins more highly abundant in astrocytes and 136 proteins more highly abundant in neurons. We performed cell type enrichment analysis of these differentially expressed proteins using protein marker lists derived from existing reference proteomes of brain cell types[5]. We found astrocytic proteins such as Gfap, Fbxo2, and Hepacam enriched in $Rosa26^{TurboID/wt}$/Aldh1l1 proteome (Fig. 5h, dark red dots) while neuronal proteins such as Syngap1, Camk2a, and Nptxr1 enriched in $Rosa26^{TurboID/wt}$/Camk2a proteome (Fig. 5h, dark blue). GSEA of the differentially expressed cell type-specific protein lists revealed enrichment of metabolic terms, including lipid metabolism, in astrocytes, while receptor (ionotropic and metabotropic), synaptic and signal transduction terms were enriched in neurons (Fig. 5i, Supplementary Data 23).

Finally, we used the adapted Luminex approach described above to measure biotinylated signaling phosphoproteins from $Rosa26^{TurboID/wt}$/Camk2a and $Rosa26^{TurboID/wt}$/Aldh1l1 brain homogenates. In comparison to $Rosa26^{TurboID/wt}$/Camk2a brain, levels of astrocyte-derived signaling phospho-proteins in the $Rosa26^{TurboID/wt}$/Aldh1l1 brains were significantly lower even after accounting for differences in efficiency of TurboID labeling across cell types (Fig. 5j, Supplementary Data 24). This suggests that basal MAPK and Akt/mTOR signaling activity is higher in Camk2a neurons compared to astrocytes under homeostatic conditions in the mouse brain.

To summarize, these findings demonstrate the ability of CIBOP to resolve proteomic differences between two brain cell types, neurons and astrocytes, and identify regional molecular differences in astrocytes at the proteomic level in their native state, without the need for isolation.

## Discussion

We report a cell type-specific in vivo proteomic labeling approach in which the biotin ligase, TurboID, is expressed in a desired cell type using a Cre-lox genetic strategy. We have generated and validated the $Rosa26^{TurboID}$ mouse line that can now be bred with a plethora of well-validated existing non-inducible and inducible Cre mouse lines to achieve tissue as well as cell type-specific proteomic labeling. The broad cellular proteomic labeling allows us to capture a snapshot of the cellular proteome of a specific cell type while retaining the native state of the cell since the labeled proteins can be purified directly from brain lysates without cell type isolation. We term this approach cell type-specific in vivo biotinylation of proteins (CIBOP). To demonstrate the utility of CIBOP, we successfully performed proof-of-principle experiments to label the neuronal proteome using AAV (AAV-CIBOP) and genetic approaches (Tg-CIBOP), providing two complementary approaches for cell type-specific labeling in vivo. The AAV-CIBOP approach can be particularly useful when using complex genetic models involving multiple transgenes, thereby avoiding need for complicated genetic crosses. However, AAV delivery does introduce some undesired effects such as glial activation and some loss of specificity. On the other hand, transgenic-CIBOP provides an ideal approach without the artefacts of AAV delivery or glial activation. Using these approaches, we then used unbiased MS and targeted immunoassays of neuron-derived (Camk2a-CIBOP) and astrocyte-derived (Aldh1l1-CIBOP) biotinylated proteins from whole brain samples. LFQ-MS allowed us to identify over 2000 proteins and regional differences within Camk2a neurons and Aldh1l1 astrocytes. A major advantage of CIBOP is its depth of proteomic coverage, even with "single-shot" label-free MS approaches. We anticipate 3-4-fold greater proteomic coverage in future studies with tandem multiplexed approaches coupled with off-line fractionation[34]. While we did not specifically perform neuron-specific phospho-proteomic profiling of neurons in our studies, we were also able to quantify over 140 neuron-derived phosphoproteins without enrichment. This depth of phosphoprotein coverage could be increased several-fold with phosphopeptide enrichment approaches[35] which could reveal cell type-specific proteomic insights at the level of post-translational modifications that cannot otherwise be captured by other methods. Our successful application of CIBOP to contrast proteomic profiles of neurons and astrocytes in their native state, revealed proteomic markers for both cell types and identified activation of phospho-protein signaling involving the MAPK and Akt/mTOR pathways as being unique to neurons as compared to astrocytes. Our studies also demonstrate the validity of this approach for broad applications in neuroscience and outside the nervous system.

Since CIBOP involves biotinylation of lysine residues, it is important to consider potential implications of excessive biotinylation on protein and cellular function. Biotin is a water soluble, readily cell permeable, and a brain penetrant vitamin (vitamin H or B7) that is required as a co-factor for several enzymes involved in glucose, amino acid, and fatty acid metabolism[36]. While biotin deficiency can cause growth retardation, skin and neurological diseases, high doses of systemically administered biotin are non-toxic[30,37]. Since TurboID is highly efficient, biotin supplementation is necessary to prevent a relative biotin deficiency caused by TurboID shunting biotin away from biotin-dependent cellular processes[21,38]. Appropriately, some concerns regarding toxicity of excessive biotinylation have been raised in the literature[21]. In the present study, we did not observe any adverse effects of neuron-specific biotinylation in $Rosa26^{TurboID}$ mice, supported by electrophysiological studies. Proteomic biotinylation also did not alter astrocyte morphology or cause a glial response in the brain. Whether shortening the duration and/or degree of biotinylation can still achieve sufficient proteomic labeling needs to be determined. As CIBOP is extended to other cell types and tissues, investigators will need to optimize the duration of Cre-mediated recombination as well as the dose and duration of biotin supplementation to individual applications.

The biotinylated neuronal proteomes from AAV- and Camk2a-CIBOP approaches were highly enriched with proteins involved in neurotransmission and axon guidance and widespread neuronal cell body, axonal, and dendritic biotinylation with no glial labeling or activation. We captured synaptic proteins, ion channels, and membrane receptors including druggable targets. AKT1, a signaling protein that is a key member of the IGF1/PI3K/AKT1/mTOR axis that is relevant to synaptic functioning and memory, is a druggable target that was identified in the AAV and Tg proteomes, highlighting consistency across both CIBOP datasets and emphasizing important roles for this pathway in neuronal physiology. At the sub-cellular level, cytosolic and membrane associated proteins such as ion channels, as well as organelles such as mitochondria, ER and Golgi apparatus, and vesicles, were abundant in the enriched neuronal proteome. We attribute the broad proteomic labeling to a combination of the biotinylation radius ($\approx$10 nm) of biotin ligases[22], the promiscuous nature of protein biotinylation by TurboID, and the use of a nuclear export sequence (NES) in the transgene of $Rosa26^{TurboID}$ mice. The low coverage of nuclear proteins observed is attributed to the NES sequence, which was chosen in order to increase the chances of labeling the extra-nuclear proteins while limiting toxicity via nuclear proteomic biotinylation. The labeling of secreted and vesicular proteins also suggests that simultaneous profiling of secreted or released proteins in tissues and biofluids may be feasible. Indeed, secretome profiling using in vivo TurboID via AAV delivery was successful in a non-brain context[39]. Our ongoing studies with $Rosa26^{TurboID}$ mice, in brain and non-brain disease contexts, will determine whether cellular origin of secreted proteins can indeed be measured in biofluids as biomarkers of underlying cellular mechanisms of disease.

The Camk2a-CIBOP transgenic strategy also successfully labeled neurons in the adult brain and, regardless of brain region, the proteome showed enrichment of neuronal synaptic proteins such as Dlg4 (PSD-95) and Gap43. With this approach, we had the opportunity to characterize regional proteomic differences within the Camk2a neurons and found core protein signatures unique to the cortex and hippocampus, striatum/thalamus, pons/medulla, and cerebellum while overcoming limitations associated with AAV-CIBOP. Glutamatergic synaptic transmission, calcium signaling, and synaptic plasticity encompassed the proteomic signature of cortical and hippocampal Camk2a neurons. Alternatively, cerebellar proteomic signatures point to mRNA processing, splicing, and calcium transporting ATPase activity. Unique proteomic signatures of pontine and medullary neurons with increased expression of metabolic proteins, inflammatory proteins, and pigmented granules agrees with the expected enrichment of axonal tracks and islands of neuronal cell bodies in the brain stem, particularly pigmented neurons[40]. Collectively, these data emphasize the diverse functions of proteins within Camk2a neurons and how each brain region may be functionally distinct from the other. Furthermore, these functionally distinct characteristics reflect neuronal-specific mechanisms within brain regions that may be vulnerable to neurological diseases.

Indeed, several protein/gene-disease associations were enriched among core regional neuronal protein lists. In cortical and hippocampal Camk2a neurons, we identified signature proteins linked to epilepsy (e.g., Dlg4) and schizophrenia (e.g., Grin2a). Several studies

have shown that the *GRIN2A* gene, which encodes a glutamate (N-methyl-D-aspartic acid [NMDA]) receptor subunit protein, contains a polymorphism in the promoter that predisposes individuals to schizophrenia[41–44]. Grin2a is also known to interact directly and indirectly with Dlg4 (PSD-95), which regulates glutamate NMDA and α-amino-3-hydroxy-5-methyl-4-isoxazoleproprionic acid (AMPA) receptor trafficking at the synapse[45] with significant roles in Alzheimer's disease (AD) biology[46]. Interestingly, regional differences in phospholipase C isoenzyme (Plcb1, Plcb4), druggable targets identified in the Camk2a proteome, expression in the brain are known to underlie several brain disorders including epilepsy, schizophrenia, and AD[47]. The neurotrophic tyrosine kinase receptor type 2 (NTRK2), a protein significantly enriched in our cortical and hippocampal Camk2a proteome associated with mood disorders, is a specific receptor for neurotrophin brain-derived neurotrophic factor (BDNF) which is implicated in the pathophysiology of bipolar disorder[48]. In the cerebellum, core signature proteins were expectedly associated with ataxias, which are defined by Purkinje cell and cerebellar circuitry dysfunction[49]. One such protein we identified is Cacna1a, a α1A-subunit of voltage-gated P/Q-type calcium channels (Cav2.1), which contains a mutation that causes spinocerebellar ataxia type 6 (SCA6) and episodic ataxia type 2[50,51]. Lastly, proteins related to inborn errors of metabolism (e.g., Acadsb)[52] and inherited polyneuropathies (e.g., Nefl) were highly abundant in the pons/medulla. Interestingly, dominant mutations in *NEFL* cause clinically distinct forms of Charcot-Marie-Tooth disease (CMT)[53–56], ranging from a severe neuropathy with an infantile onset, to a more moderate neuropathy with onset typically between 10 and 20 years. In summary, these regional protein/gene-disease associations demonstrate the ability of CIBOP to identify mechanisms that underlie selective neuronal and regional vulnerability in various neurological diseases[57].

We extended the Tg-CIBOP strategy to characterize the proteome of non-neuronal cells, specifically astrocytes, by crossing *Rosa26*[TurboID] mice to Aldh1l1-Cre[Ert2] mice. Similar to Camk2a-CIBOP, Aldh1l1-CIBOP labeled the astrocyte proteome with comparable efficiency across different brain regions. The biotinylated proteins appeared to be within astrocyte cell bodies and along blood vessels, which is expected given that astrocytic end-feet contact blood vessels to form the glia limitans of the blood brain barrier[58]. The astrocyte proteome showed enrichment of metabolic (e.g., Aldoc), glial (e.g., Gfap), and lipid binding (e.g., Lrp1) proteins. In addition, we identified synaptic terms in the astrocyte proteome. One explanation for this unexpected finding is the synaptic proteins were potentially phagocytosed by astrocytes since they actively contribute to synapse pruning and elimination[59]. Complementary to Camk2a-CIBOP, we characterized the regional proteomic differences within the astrocytes and found core protein signatures unique to the cortex and hippocampus, pons/medulla, cerebellum, and spinal cord. Sodium ion transport, biotin carboxylase activity, and pyruvate metabolic process encompassed the proteomic signature of cerebellar astrocytes while terms such as glutamate receptor activity, long term potentiation, and ion channel complex dominated the cortical and hippocampal astrocyte proteome. Alternatively, pontine and medullary proteomic signatures point to microtubule binding and regulation of neurotransmitter transport. Extracellular matrix, collagen, and protein heterodimerization were unique to the spinal cord. We were also able to distinguish cell type-specific proteomes by comparing astrocyte and neuronal proteomes. As expected, processes such as lipid metabolism, solute transport, mitochondrion, and metabolic activities were unique to the astrocyte proteome while synaptic transmission and long-term potentiation were relatively enriched in the neuronal proteome. Collectively, these data highlight the various roles of astrocytes within mouse brain regions under native states and how their proteome is distinct from neurons.

Using the adapted Luminex approach to directly measure biotinylated proteins of interest from brain homogenates, we found that several MAPK (e.g., pERK1/2 and pMEK1/2) and Akt/mTOR (e.g., pGSK3 α and β) signaling phosphoproteins are highly abundant in neurons while majority of cytokines in the brain, with the exceptions of IL-10, IL-2, TNF-α, IL-1α, and RANTES were mostly of non-neuronal origin. Previous studies have shown that neurons may produce cytokines such as IL-2 and TNF-α under homeostatic conditions[60–63]. We also resolved differences in neuronal MAPK and Akt/mTOR signaling activation across brain regions, with cortical Camk2a neurons exhibiting the highest level of baseline activation Akt/mTOR signaling and striatal/thalamic neurons exhibiting highest level of MAPK. From a disease standpoint, recent studies have shown that increased MAPK protein expression and MAPK pathway activation (e.g., ERK signaling) are highly characteristic and potentially causal, in neurodegenerative diseases such as AD[64,65], where they are associated with cognitive decline and pathological burden. MAPK signaling is also critical for synaptic mechanisms such as long-term potentiation and response to stress or injury[64]. Akt signaling activates the mTOR pathway that regulates mRNA translation, metabolism, and protein turnover, and is highly relevant to the pathogenesis of lysosomal storage disorders, genetic neurodevelopmental disorders, and neurodegenerative disorders[66]. The ability of CIBOP to label and quantify neuron-specific MAPK and Akt signals from whole brain can be highly relevant to understanding the spatio-temporal dynamics of these signaling pathways in disease pathogenesis. While our observed regional differences in neuron-derived cytokines are exploratory, the inverse correlation between MAPK and Akt signaling (high in the cortex but low in cerebellum) and cytokine levels (low in the cortex but high in the cerebellum), warrant further investigation[67]. Using the adapted Luminex approach to contrast neurons and astrocytes, we also found that neurons exhibit higher levels of basal signaling activity involving the MAPK and Akt pathways as compared to astrocytes, validating our mass spectrometry findings of higher levels of signaling proteins in neurons. The results presented in this manuscript also lay the foundation for future studies to determine whether neuroinflammatory stimuli, neuropathology and injury can alter signaling pathway activation and cytokine production in neurons and glia with spatial resolution.

In conclusion, we have generated and validated the *Rosa26*[TurboID] mouse line that enables CIBOP, a powerful experimental approach to investigate molecular changes occurring at the proteomic level while retaining the native state of the cell. Data obtained using this versatile in vivo system showcases breadth of proteomic labeling in neurons and astrocytes, regional Camk2a neuronal and astrocyte proteomic signatures, and neuron and astrocyte-specific phosphoprotein signaling (MAPK and Akt/mTOR) and cytokine signatures with relevance to several neurological disorders. These native state Camk2a neuronal and astrocyte proteomes represent key resources for the neuroscience community. This *Rosa26*[TurboID] mouse model and our validated experimental workflow also provide a framework to resolve cellular mechanisms underlying physiological or pathological conditions of complex tissues.

## Methods

**Construct generation and cell culture studies.** The V5-TurboID-NES_pCDNA3, a gift from Alice Ting[21] (Addgene plasmid # 107169) was used to generate AsiS1-Kozak-V5-TurboID-NES-stop-Mlu1 construct. This was then cloned into the pR26 CAG AsiSI/MluI targeting vector, a gift from Ralf Kuehn[68] (Addgene plasmid # 74286), to generate the a *Rosa26* (chromosome 6) targeting vector containing the CAG promoter, a floxed STOP site (loxp-STOP-loxp), and V5-TurboID with a nuclear export signal (TurboID-NES). This *Rosa26*[TurboID] targeting vector was verified in-vitro for Cre-mediated TurboID expression and biotinylation in HEK293 cells (Supplementary Fig. 1a, b).

Human embryonic kidney 293 (HEK293 from ATCC CRL-1573) cells were maintained in Dulbecco's modified Eagle's medium (DMEM) supplemented with 10% fetal bovine serum (FBS) and 1% penicillin/streptomycin at 37 °C in a 5% $CO_2$ atmosphere. For transient transfection, cells grown to 70–80% confluency in 6-well plates were transfected (Lipofectamine 3000, Thermo, L3000001) with 2.5 μg/well of $Rosa26^{TurboID}$ targeting vector and Cre plasmid (CMV-Cre, a gift from Dr. Xinping Huang, Emory University), $Rosa26^{TurboID}$ targeting vector alone, or Cre plasmid alone according to manufacturer's protocol. Untransfected cells also served as negative controls. Twenty-four hours post-transfection, cells were treated with 200 μM biotin for another 24 h. Subsequently, cells were rinsed with cold 1X phosphate buffered saline (PBS) and harvested in urea lysis buffer (8 M urea, 10 mM Tris, 100 mM $NaH_2PO_4$, pH 8.5) containing 1X HALT protease inhibitor cocktail without EDTA (Thermo, 87786). The cells were sonicated for 3 rounds consisting of 5 s of active sonication at 25% amplitude with 10 s incubation periods on ice between sonication. Lysed cells were then centrifuged for 5 min at 15,000 × g and the supernatants were transferred to a new tube. Protein concentration was determined by bicinchoninic acid (BCA) assay (Thermo, 23225). To confirm protein biotinylation, 10 μg of cell lysates were resolved on a 4–12% Bris-Tris gel, transferred onto a nitrocellulose membrane, and probed with streptavidin-Alexa680 (Thermo, S32358) diluted 1:10 K in Start Block (Thermo, 37543) for 1 h at room temperature. Subsequently, the membrane was washed in 1X tris buffered saline containing 0.1% Tween20 (TBS-T) and incubated over-night with mouse anti-V5 (1:250; Thermo, R960-25) or anti-Gapdh (1:3000; Abcam, ab8245) diluted in Start Block. After washes in TBS-T, membranes were incubated with goat anti-mouse 700 (1:10 K) and imaged using Odyssey Infrared Imaging System (LI-COR Biosciences).

For immunocytochemistry (ICC), cells grown to 50% confluency on coverslips were transfected and treated with biotin as described above. Subsequently, cells were washed 3X with warm sterile PBS for 5 min each and fixed with 1X ICC fixation buffer (Invitrogen, 00-8222-49) for 30 min. After PBS washes, cells were permeabilized (Invitrogen, 00-8333-56) for 20 min and blocked in 10% normal horse serum (NHS) in PBS for 45 min at room temperature. The cells were then incubated with mouse anti-V5 diluted in 2% NHS in PBS overnight. After thorough washes with PBS, cells were incubated with anti-mouse Rhodamine Red (1:500, goat anti-mouse Rhodamine-red, Thermo, R6393) and streptavidin Alexa-flour 488 (1:500, Thermo, S11223) diluted in 2% NHS in PBS for 1 h. Coverslips were mounted onto slides using mounting media containing DAPI (Sigma-Aldrich, F6057) and images were captured using an Olympus fluorescence microscope (Olympus BX51) and camera (Olympus DP70) and processed using Image J software (FIJI Version 1.51).

**Generation of the $Rosa26^{TurboID}$ mouse.** The $Rosa26^{TurboID}$ targeting vector was electroporated into C57BL/6 N embryonic stem (ES) cells (from Taconic, ES Cell Line # JM8A3) at Texas A&M Institute for Genomic Medicine (TIGM). After confirming homologous recombination in ES clones, they were microinjected into albino goGermline™ blastocysts (Ozgene) and implanted into pseudo pregnant CD-1 female mice[21] (Emory Mouse Transgenic and Gene Targeting Core). The resulting chimeric mice were crossed to wild-type (WT) C57BL6/J[69] mice to yield F1 $Rosa26^{TurboID}$ heterozygous mice ($Rosa26^{TurboID/wt}$) and littermate controls. Genotyping was performed on DNA extracted from a tail biopsy from mice using the following primers: (TurboID_fwd) 5' ATCCCGCTGCTGAACGCTAAAC 3', (TurboID_rev) 5' ACCATTTCCTCCCTCTGCTTCC 3', (ROSA26_fwd) 5' CTCT TCCCTCGTGATCTGCAACTCC 3', (ROSA_rev) 5' CATGTCTTTAATCTAC CTCGATGG 3'. Mice were designated as heterozygous ($Rosa26^{TurboID/wt}$) by the presence of a band approximately 181 bp corresponding to the TurboID transgene and a 299 bp band produced by the endogenous ROSA26 allele, while WT contained only the 299 bp endogenous ROSA26 band (Supplementary Fig. 1c).

**Animal studies.** Approval from the Emory University Institutional Animal Care and Use Committee was obtained prior to all animal-related studies (IACUC protocols # PROTO201800252 and PROTO201700821). All mice used in the present study were housed in the Department of Animal Resources at Emory University under a 12 h light/12 h dark cycle with ad libitum access to food and water. Animals were housed in the vivarium under standard conditions for mice (temperature 72 F, humidity range 40–50%). All procedures were approved by the Institutional Animal Care and Use Committee of Emory University and were in strict accordance with the National Institute of Health's "Guide for the Care and Use of Laboratory Animals".

AAV approach for Cre recombinase delivery to neurons in $Rosa26^{TurboID/wt}$ mice: 2-month-old mice were anesthetized with isoflurane, given sustained-release buprenorphine subcutaneously (0.5 mg/kg), and immobilized on a stereotaxic apparatus. The mice were maintained on 1–2.5% isoflurane and monitored closely for breathing abnormalities throughout the surgery. Bilateral intrahippocampal injections (coordinates from bregma: −2.1 mm posterior, ±2.0 mm lateral, and ±1.8 mm ventral) were performed over a 5 min period with 1 μL of AAV9-hSyn-Cre (Titer ≥ 1 × 10^{13} vg/mL, Addgene, 105553-AAV9) or un-injected: un-injected WT mice (n = 3, male), wild-type mice injected with AAV9-hSyn-Cre (n = 3, male), and $Rosa26^{TurboID/wt}$ mice injected with AAV9-hSyn-Cre (n = 3, male) (Fig. 1b). The incision was closed with tissue adhesive (Fisher, NC0304169), isoflurane was discontinued, and the animal was revived in a new, clean cage atop a

heating pad. The mice were monitored every 15 min for 1 h after surgery and routinely for the 3-day post-surgery survival period under normal vivarium conditions. After 4 weeks, mice were given water supplemented with biotin[30] (37.5 mg/L)[30] for 2 weeks until euthanasia at 3 months of age.

Transgenic approach for Cre recombinase expression in neurons and astrocytes using $Rosa26^{TurboID/wt}$ mice:

Cohort 1: $Rosa26^{TurboID/wt}$ mice were crossed with Camk2a-Cre^{Ert2} mice (Jackson Labs, Stock No. 012362) to obtain heterozygous Camk2a-Cre^{Ert2} ("Camk2a", n = 2; male) and $Rosa26^{TurboID/wt}$/Camk2a-Cre^{Ert2} ("$Rosa26^{TurboID/wt}$/Camk2a", n = 2; 1 male and 1 female) littermate mice. All mice were given tamoxifen (75 mg/kg) intraperitoneally for 5 days at 6 weeks of age. After 3 weeks, mice were given water supplemented with biotin (37.5 mg/L)[30] for 2 weeks until euthanasia at 3 months of age. Consistent with previous publications[27], we have previously validated the Camk2a-cre^{Ert2} line for neuron-specific labeling and non-leaky Cre activity.

Cohort 2: $Rosa26^{TurboID/wt}$ mice were crossed with; to obtain heterozygous $Rosa26^{TurboID/wt}$/Aldh1l1-Cre^{Ert2} ("$Rosa26^{TurboID/wt}$/Aldh1l1", n = 2; 1 male and 1 female). We also generated $Rosa26^{TurboID/wt}$/Camk2a (n = 2; male) mice as described above. Heterozygous $Rosa26^{TurboID/wt}$ littermate mice (n = 3; male) were used as controls. The same tamoxifen and biotin water supplementation paradigms were followed as described above.

After biotin supplementation (37.5 mg/L) for 2 weeks, mice were anesthetized with ketamine (ketamine 87.5 mg/kg, xylazine 12.5 mg/kg) followed by transcardial perfusion with 30 mL of ice-cold PBS. The brain was immediately removed and hemi-sected along the mid-sagittal line. The left hemisphere was fixed in 4% paraformaldehyde (PFA) for 24 h and then transferred to 30% sucrose after through washes in PBS. The right hemisphere was either immediately snap frozen on dry ice for the AAV cohort or dissected for the following brain regions for the transgenic cohort and then snap frozen: cortex (CTX), hippocampus (HIP), cerebellum (CB), pons/medulla (PM), and striatum/thalamus (ST) (Fig. 2a). We also collected the entire spinal cord for transgenic cohort 2 (Fig. 5a).

**Tissue homogenization and immunoblotting.** Frozen brain tissue pieces (AAV cohort: forebrain without CB and PM; transgenic cohorts: dissected brain regions) were added to 1.5 mL Rino tubes (Next Advance) containing stainless-steel beads (0.9–2 mm in diameter) and five volumes of the tissue weight in urea lysis buffer (8 M urea, 10 mM Tris, 100 mM $NaH_2PO_4$, pH 8.5) containing 1X HALT protease inhibitor cocktail without EDTA. Tissues were homogenized in a Bullet Blender (Next Advance) twice for 5 min cycles at 4 °C. The homogenates were transferred to a new Eppendorf LoBind tube followed by 3 rounds of sonication consisting of 5 s of active sonication at 25% amplitude with 10 s incubation periods on ice between sonication. Homogenates were then centrifuged for 5 min at 15,000 × g and the supernatants were transferred to a new tube. Protein concentration was determined by BCA assay.

To confirm protein biotinylation, 20 μg of brain lysates were resolved on a 4–12% Bris-Tris gel, transferred onto a nitrocellulose membrane, and probed with streptavidin-Alexa 680 diluted in Start Block for 1 h at room temperature. Subsequently, the membrane was incubated over-night with mouse anti-V5 or anti-Gapdh diluted in Start Block. After washes in TBS-T, membranes were incubated with goat anti-mouse 700 (1:10 K) and imaged using Odyssey Infrared Imaging System (LI-COR Biosciences). Biotinylation was quantified using Odyssey Studio Lite. For densitometry, a box around the entire length of streptavidin signal was drawn for every lane and normalized to Gapdh signal.

**Biotinylated protein enrichment.** After confirmation of biotinylation of proteins and V5 expression by Western blot, biotin-tagged proteins were enriched using a previously published protocol[22] with slight modifications. Briefly, for each sample, 83 μL of streptavidin magnetic beads (Thermo, 88817) in a 1.5 mL Eppendorf LoBind tube were washed twice with 1 mL RIPA lysis buffer (50 mM Tris, 150 mM NaCl, 0.1% SDS, 0.5% sodium deoxycholate, 1% Triton X-100) on rotation for 2 min. The beads were incubated at 4 °C for at least 1 h with rotation with 500 μg - 1 mg protein from each sample with an addition 500 μL RIPA lysis buffer. The beads were centrifuged briefly, placed on a magnetic rack, and the supernatant was collected into a new 1.5 mL Eppendorf LoBind tube and frozen at −20 °C. The beads were washed in the following series of buffers on rotation at room temperature: twice with 1 mL RIPA lysis buffer for 8 min, once with 1 mL 1 M KCl for 8 min, once with 1 mL 0.1 M sodium carbonate ($Na_2CO_3$) for ~10 s, once with 1 mL 2 M urea in 10 mM Tris-HCl (pH 8.0) for ~10 s, and twice with 1 mL RIPA lysis buffer for 8 min. The beads, in the final RIPA lysis buffer wash, were transferred to a new tube and washed twice with 1 mL of PBS for 2 min on rotation. After the final wash, beads were resuspended in 83 μL of PBS. To verify protein enrichment, 10% of the bead volume was transferred to a new tube and were boiled in 30 μL of 2X protein loading buffer (Biorad, 1610737) supplemented with 2 mM biotin and 20 mM dithiothreitol (DTT) at 95 °C for 10 min to elute the biotinylated proteins. Subsequently, 10 μL of eluate was run on a gel and probed with Streptavidin680 while 20 μL of the eluate was run on a separate gel and Silver stained (Thermo, 24612). All samples in the $Rosa26^{TurboID/wt}$ AAV cohort (n = 3/condition, $n_{total}$ = 9) were enriched as described above. For the transgenic cohort 1 (n = 2/brain region/genotype), we enriched from 500 μg of protein and prepared technical replicates for enrichment from the $Rosa26^{TurboID/wt}$/Camk2a brain

regions (excluding hippocampus), which resulted in $n = 4$/region and $n_{total} = 18$, while the control Camk2a brain regions did not contain technical replicates (Fig. 2a). Similarly, ($n = 2$/brain region/genotype), we enriched from 500 µg of protein and prepared technical replicates for enrichment from the *Rosa26-*$^{TurboID/wt}$/Camk2a ($n_{total} = 22$) and *Rosa26*$^{TurboID/wt}$/Aldh1l1 ($n_{total} = 22$) brain regions (excluding hippocampus), including the spinal cord, while the control *Rosa26*$^{TurboID/wt}$ brain regions did not contain technical replicates.

**Protein digestion.** To prepare enriched samples for mass spectrometry analysis, the remaining 90% of streptavidin beads (described above) were washed three times with PBS and then resuspended in 50 mM ammonium bicarbonate ($NH_4HCO_3$). Bound proteins were then reduced with 1 mM dithiothreitol (DTT) at room temperature for 30 min and alkylated with 5 mM iodoacetamide (IAA) in the dark for 30 min with rotation. Proteins were digested overnight with 0.5 µg of lysyl (Lys-C) endopeptidase (Wako, 127-06621) at RT on shaker followed by further overnight digestion with 1 µg trypsin (Thermo, 90058) at RT on shaker. The resulting peptide solutions were acidified to a final concentration of 1% formic acid (FA) and 0.1% triflouroacetic acid (TFA), desalted with a HLB columns (Waters, 186003908), and dried down in a vacuum centrifuge (SpeedVac Vacuum Concentrator).

To prepare brain homogenates, 50 µg of protein from pooled total brain lysates from the AAV cohort were reduced with 5 mM DTT at room temperature for 30 min and alkylated by 10 mM IAA in the dark for 30 min. Samples were then diluted (4-fold) with 50 mM ammonium bicarbonate (ABC) and digested overnight with 1 µg of Lys-C at RT on shaker. Samples were further diluted 4-fold with 50 mM ABC and digested overnight with 2 µg of trypsin at RT on shaker. The peptide solutions were acidified to a final concentration of 1% FA and 0.1% TFA, desalted with HLB columns, and dried down in a vacuum centrifuge (SpeedVac Vacuum Concentrator). Similarly, 50 µg of protein from brain region homogenates (not-pooled) from transgenic cohort 1 (Camk2a-Cre$^{Ert2}$ and *Rosa26*$^{TurboID/wt}$/Camk2a-Cre$^{Ert2}$) were digested and desalted.

**Mass spectrometry.** Dried peptides were resuspended in 15 µL of loading buffer (0.1% FA and 0.03% TFA in water), and 7–8 µL was loaded onto a self-packed 25 cm (100 µm internal diameter packed with 1.7 µm Water's CSH beads) using an Easy-nLC 1200 or Dionex 3000 RSLCnano liquid chromatography system. The liquid chromatography gradient started at 1% buffer B (80% acetonitrile with 0.1% FA) and ramps to 5% in 10 s. This was followed by a 55 min linear gradient to 35% B and finally a 4 min 50 s 99% B flush. For the AAV cohort (IP and total brain samples), an Orbitrap Lumos Tribrid mass spectrometer with a high-field asymmetric waveform ion mobility spectrometry (FAIMS Pro)[70] interface was used to acquire all mass spectra at a compensation voltage of −45V. The spectrometer was operated in data dependent mode in top speed mode with a cycle time of 3 s. Survey scans were collected in the Orbitrap with a 60,000 resolution, 400 to 1600 m/z range, 400,000 automatic gain control (AGC), 50 ms max injection time and rf lens at 30%. Higher energy collision dissociation (HCD) tandem mass spectra were collected in the ion trap with a collision energy of 35%, an isolation width of 1.6 m/z, AGC target of 10000, and a max injection time of 35 ms. Dynamic exclusion was set to 30 s with a 10 ppm mass tolerance window.

For transgenic brain region samples (IP and total brain samples), an Orbitrap Eclipse Tribrid mass spectrometer with a high-field asymmetric waveform ion mobility spectrometry (FAIMS Pro)[70] interface was used to acquire all mass spectra at a compensation voltage of −45V. The spectrometer was operated in data dependent mode in top speed mode with a cycle time of 3 s. Survey scans were collected in the Orbitrap with a 120,000 resolution, 400 to 1600 m/z range, 400,000 automatic gain control (AGC), 50 ms max injection time and rf lens at 30%. Higher energy collision dissociation (HCD) tandem mass spectra were collected in the ion trap with a collision energy of 35%, an isolation width of 0.7 m/z, AGC target of 10000, and a max injection time of 35 ms. Dynamic exclusion was set to 30 s with a 10 ppm mass tolerance window. The mass spectrometry proteomics data have been deposited to the ProteomeXchange Consortium via the PRIDE[71] partner repository with the dataset identifiers PXD027488 and PXD032161.

**Protein identification and quantification.** MS raw files were searched using the search engine Andromeda, integrated into MaxQuant, against 2020 mouse Uniprot database (91,441 target sequences including peptide sequences for V5 and TurboID). Methionine oxidation (+15.9949 Da) and protein N-terminal acetylation (+42.0106 Da) were variable modifications (up to 5 allowed per peptide); cysteine was assigned as a fixed carbamidomethyl modification (+57.0215 Da). Only fully tryptic peptides were considered with up to 2 missed cleavages in the database search. A precursor mass tolerance of ±20 ppm was applied prior to mass accuracy calibration and ±4.5 ppm after internal MaxQuant calibration. Other search settings included a maximum peptide mass of 4600 Da, a minimum peptide length of 6 residues, 0.05 Da tolerance for orbitrap and 0.6 Da tolerance for ion trap MS/MS scans. The false discovery rate (FDR) for peptide spectral matches, proteins, and site decoy fraction were all set to 1 percent. Quantification settings were as follows: re-quantify with a second peak finding attempt after protein identification has completed; match MS1 peaks between runs; a 0.7 min retention time match window was used after an alignment function was found with a 20 min RT search

space. Quantitation of proteins was performed using summed peptide intensities given by MaxQuant. The quantitation method only considered razor plus unique peptides for protein level quantitation.

The MaxQuant output data were uploaded onto Perseus (Version 1.6.15) for analyses. The categorical variables were removed, and intensity values were log (base 2) transformed. The data were filtered to contain at least three valid values across nine samples to accommodate six missing values in the negative controls for the AAV cohort and missing values were further imputed from normal distribution (width: 0.3, down shift: 1.8) (Supplementary Data 1). The MaxQuant output data from pooled total brain lysates from the AAV cohort were filtered to contain at least 1 valid value across three samples and missing samples were imputed from normal distribution (width: 0.3, down shift: 1.8). The MaxQuant output data from the transgenic cohort 1 (*Rosa26*$^{TurboID/wt}$/Camk2a only) brain regions were first filtered to contain at least nine valid intensity values in the transgenic IP samples and then processed as described above with Perseus (Supplementary Data 9). Lastly, the MaxQuant output data from the transgenic cohort 2 brain regions were first filtered to contain at least 11 valid intensity values in each group, *Rosa26*$^{TurboID/wt}$/Camk2a and *Rosa26*$^{TurboID/wt}$/Aldh1l1, and then processed as described above with Perseus (Supplementary Data 19). Afterwards, the abundance values in the transgenic IP samples were normalized to TurboID abundance across all brain regions.

**Differential expression, Gene Ontology analysis, Clustering, and data visualization.** We applied differential expression, gene ontology (GO) analysis, as well as principal component analysis (PCA) and hierarchical clustering approaches (average linkage method, one minus Pearson correlation) to analyze proteomic data from both *Rosa26*$^{TurboID/wt}$ AAV and transgenic cohorts. Heat maps of normalized data were generated using Morpheus (Broad Institute, Morpheus, https://software.broadinstitute.org/morpheus) and additional graphical representation of data was performed using SPSS (IBM, Version 26.0), R software (R-4.2.0), and Prism (Graphpad, Version 9).

For the AAV cohort, significantly differentially enriched proteins (unadjusted $p$ value ≤ 0.05) were identified by an unpaired t-test comparing a) *Rosa26*$^{TurboID/wt}$/hSyn vs. both control groups, b) *Rosa26*$^{TurboID/wt}$/hSyn vs. WT un-injected, or c) *Rosa26*$^{TurboID/wt}$/hSyn vs. WT/hSyn (Supplementary Data 1). Differentially enriched proteins are presented as volcano plots.

For the transgenic cohorts, prior to any analyses, protein abundances in *Rosa26*$^{TurboID/wt}$/Camk2a (cohort 1) or *Rosa26*$^{TurboID/wt}$/Camk2a and *Rosa26*$^{TurboID/wt}$/Aldh1l1 (cohort 2) brain regions were normalized to TurboID levels to adjust for regional variability. Then, we used an unpaired t-test comparing all brain regions from controls vs. all brain regions from labeled mice to identify differentially enriched or depleted proteins, shown as a volcano plot (Supplementary Data 9 (cohort 1), Supplementary Data 19 (cohort 2)). To identify core enriched proteins across brain regions of *Rosa26*$^{TurboID/wt}$/Camk2a mice and *Rosa26*$^{TurboID/wt}$/Aldh1l1, we performed an unpaired t-test comparing one brain region vs. all other brain regions (i.e., PM vs. [CTX, HIP, ST, CB]) and displayed proteins that were significantly ($p ≤ 0.05$) ≥4 fold enriched in a heatmap (Supplementary Data 11 (cohort 1), Supplementary Data 21 (cohort 2)).

Gene set enrichment analysis (GSEA) of differentially enriched proteins was performed using AltAnalyze (Version 2.0). GO terms meeting Fisher exact significance of $p$ value ≤ 0.05 (i.e., a Z-score greater than 1.96) were considered. Input lists included proteins that were significantly differentially enriched (unadjusted $p ≤ 0.05$, ≥2 fold) from the various differential analyses described above (Supplementary Data 3, 7, 8, 10, 20, 23). The background gene list consisted of unique gene symbols for all proteins identified and quantified in the mouse brain ($n = 7736$)[5]. For the regional analysis of neuronal proteomic data, we identified proteins that were at least 4-fold enriched in the specific region compared to all other brain regions, representing a core set of regionally enriched proteins. These regional protein lists were searched against curated sets of gene-disease, gene-trait, and gene-phenotype associations (DisGenNet v7.0) and the results relevant to neurological or psychiatric traits were sorted based on gene-disease association scores[72] (Supplementary Data 13). Networks of known direct (protein-protein) and indirect (functional) interactions within core regional protein signatures were explored using publicly available sources of these interactions (STRING Consortium 2020 Version 11.0)[73]. In order to identify druggable protein targets, we searched our AAV and transgenic proteomic data against a list ($n = 1326$) of potential drug targets that belong to the following drug target protein classes: enzymes, transporters, receptors, and ion-channels (https://www.proteinatlas.org/humanproteome/tissue/druggable).

**Immunofluorescent staining and microscopy.** Fixed brains were cut into 30 µm thick sagittal sections using a cryostat. For immunofluorescence (IF) staining, 2-3 brain sections from each mouse were thoroughly washed to remove cryopreservative, blocked in 8% normal horse serum (NHS), diluted in TBS containing 0.1% Triton X-100 for 1 h, and incubated with primary antibodies diluted in PBS containing 2% NHS overnight (1:500 rabbit anti-Iba1, Abcam, ab178846; 1:200 mouse anti-Map2, BD Pharmagen, 556320; 1:500 mouse anti-Gfap, Thermo, 14-9892-82). Following thorough washes and incubation in the appropriate fluorophore-conjugated secondary antibody (1:500, goat anti-mouse Rhodamine-red, Thermo, R6393; or anti-rabbit Rhodamine-red Thermo, R6394) and

streptavidin Alexa-flour 488 (1:500, Thermo, S11223) for 1 h at room temperature, sections were mounted on slides with mounting media containing DAPI (Sigma-Aldrich, F6057) for nuclear staining. Representative images of the same regions across all samples were taken using the Keyence BZ-X810 and all image processing was performed using Image J software (FIJI Version 1.51).

**Confocal microscopy.** 30 μm thick free-floating brain sections were washed, blocked and permeabilized by incubating in TBS containing 0.25% Triton X-100 and 5% horse serum for 1 h at room temperature. Primary antibodies (1:500 rabbit anti-Iba1, Abcam, ab178846; 1:500 mouse anti-βIII tubulin, Promega, G712A; 1:500 mouse anti-GFAP, Millipore, MAB360; 1:500 rabbit anti-ndrg2, Thermofisher, PA5-79722; rabbit anti-AQP4, Sigma, HPA014784) were diluted in TBS containing 0.25% Triton TX-100 and 1% horse serum. After overnight incubation at 4 °C with primary antibodies, the sections were rinsed 3x in TBS containing 1% horse serum at room temperature for 10 min each. Then, the sections were incubated in the appropriate fluorophore-conjugated secondary antibody (1:500; donkey anti-mouse Alexa Fluor 594, Thermofisher, A21203; donkey anti-rabbit Alexa Fluor 594, Thermofisher, A-21207; donkey anti-rabbit Alexa Fluor plus 647, Thermofisher, A32795) and streptavidin Alexa-flour 488 at room temperature for 2 h in dark. The sections were rinsed once and incubated with DAPI for 10 min, washed 3x in TBS for 10 min, dried, and cover slipped with ProLong Diamond Antifade Mountant (ThermoFisher, P36965). Sections were imaged as z-stacks on a Nikon A1R HD25 inverted confocal microscope with a 40X objective (NA 1.3) using NIS-Elements Imaging software. Images were processed and maximum intensity projections were created using Image J.

**Quantification of cytokines and signaling phospho-protein levels in brain homogenates.** Luminex multiplexed immunoassays were used to quantify phospho-proteins within the MAPK (Millipore 48-660MAG) and PI3K/Akt/mTOR (Millipore 48-612MAG) pathways as well as 32 panel of cytokine/chemokine protein expression (Millipore MCYTMAG-70K-PX32) in brain lysates from the *Rosa26^TurboID/wt* AAV and transgenic cohorts. The cytokines panel detected: Eotaxin/CCL11, G-CSF, GM-CSF, IFN-γ, IL-1α, IL-1β, IL-2, IL-3, IL-4, IL-5, IL-6, IL-7, IL-9, IL-10, IL-12 (p40), IL-12 (p70), IL-13, IL-15, IL-17, IP-10, KC, LIF, LIX, MCP-1, M-CSF, MIG, MIP-1α, MIP-1β, MIP-2, RANTES, TNF-α, VEGF. The PI3K/Akt/mTOR panel detected: pGSK3α (Ser21), pIGF1R (Tyr1135/Tyr1136), pIRS1 (Ser636), pAkt (Ser473), p-mTOR (Ser2448), p70S6K (Thr412), pIR (Tyr1162/Tyr1163), pPTEN (Ser380), pGSK3β (Ser9), pTSC2 (Ser939), RPS6 (Ser235/Ser236). The MAPK panel detected: pATF2 (Thr71), pErk (Thr185/Tyr187), pHSP27 (Ser78), pJNK (Thr183/Tyr185), p-c-Jun (Ser73), pMEK1 (Ser222), pMSK1 (Ser212), p38 (Thr180/Tyr182), p53 (Ser15), pSTAT1 (Tyr701). Assays were read on a MAGPIX instrument (Luminex).

Luminex assays consist of multiplexed sandwich ELISAs occurring on magnetic beads, with distinct fluorescent labels for each analyze quantified. Samples were analyzed using both "standard" and "adapted" Luminex assays (Fig. 4a). In the standard assay, beads were first incubated with sample, followed by biotinylated detection antibody, then streptavidin-phycoerythrin, which provides the fluorescent intensity readout (Fig. 4a). The adapted assay takes advantage of the cell type-specific biotinylated proteome and omits the biotinylated detection antibody step (Fig. 4a). Linear range analysis was conducted to identify a range of total protein loaded that corresponded to linear signal readout from the Luminex instrument for each analyte. Based on this, 6 μg, 0.375 μg, and 2 μg of total protein from the AAV cohort was loaded for the cytokine, PI3K/Akt/mTOR, and MAPK assays, respectively. For the brain regions from the transgenic cohort 1, 2.5 μg, 2 μg, and 0.75 μg of total protein was loaded for the cytokine, PI3K/Akt/mTOR, and MAPK assays, respectively. For the brain regions from the transgenic cohort 2, 1.5 μg, 1.5 μg, and 0.5 μg of total protein was loaded for the cytokine, PI3K/Akt/mTOR, and MAPK assays, respectively

For the *Rosa26^TurboID/wt* AAV cohort, the background was subtracted from fluorescence intensity values for each analyte from the standard and adapted assays. Data from each assay are shown as bar graphs and are detailed in Supplementary Data 17. For the brain regions from the *Rosa26^TurboID/wt* transgenic cohorts, fluorescent intensity values from the adapted Luminex assay from the control mouse samples were subtracted from the intensity values from the adapted assay from *Rosa26^TurboID/wt*/Camk2a or *Rosa26^TurboID/wt*/Aldh1ll1 mouse samples. These values were then normalized to TurboID abundance from mass spectrometry intensity for each region to account for differences in promoter activity and TurboID protein expression. We then performed hierarchical clustering analysis of the normalized data (Morpheus) to identify regional patterns which were visualized as heatmaps (raw and normalized data included in Supplementary Data 18 and 24).

To determine whether the measured signal in the adapted assay was due to biotinylation of proteins, we performed blocking experiments where samples were incubated with beads to capture proteins of interest, then incubated with monomeric avidin (0.05% m/V) for 1 h to block proteins biotinylated by TurboID, and then followed by 1 h blocking with biotin (0.05% m/V) to quench any available avidin binding sites. This was followed by addition streptavidin fluorophore while skipping the detection antibody, as described above in the adapted assay. These were performed on samples from Camk2a controls and *Rosa26^TurboID/wt*/Camk2a brain lysates (Supplementary Fig. 8).

**Acute hippocampal slice preparation.** Acute hippocampal slices were prepared from 3-month-old *Rosa26^TurboID/wt*/Camk2a mice or littermate control *Rosa26^TurboID/wt* after receiving tamoxifen and biotin supplementation. Mice were first anesthetized and perfused with ice-cold cutting solution (in mM) 87 NaCl, 25 NaHO$_3$, 2.5 KCl, 1.25 NaH$_2$PO$_4$, 7 MgCl$_2$, 0.5 CaCl$_2$, 10 glucose, and 7 sucrose. Thereafter, the brain was immediately removed by dissection. Brain slices (300 μm) were sectioned in the coronal plane using a vibrating blade microtome (Leica Biosystems, VT1200S) in the same solution. Slices were transferred to an incubation chamber and maintained at 34 °C for 30 min and then at room temperature (23–25 °C). During whole-cell recordings, slices were continuously perfused with (in mM) 128 NaCl, 26.2 NaHO$_3$, 2.5 KCl, 1 NaH$_2$PO$_4$, 1.5 CaCl$_2$, 1.5 MgCl$_2$, and 11 glucose, maintained at 30.0 ± 0.5 °C. All solutions were equilibrated and maintained with carbogen gas (95% O$_2$/5% CO$_2$) throughout.

**Electrophysiology.** Pyramidal neurons were targeted for somatic whole-cell recording in the CA3c region of hippocampus using gradient-contrast video-microscopy on custom-built or commercial (Bruker) upright microscopes. This region was selected based on high level of Camk2a expression in these neurons[32]. Electrophysiological recordings were obtained using Multiclamp 700B amplifiers (Molecular Devices). Signals were filtered at 10 kHz and sampled at 50 kHz using a Digidata 1440B (Molecular Devices) digitizer. For whole-cell recordings, borosilicate patch pipettes were filled with an intracellular solution containing (in mM) 124 potassium gluconate, 2 KCl, 9 HEPES, 4 MgCl$_2$, 4 NaATP, 3 L-Ascorbic Acid, and 0.5 NaGTP. Pipette capacitance was neutralized in all recordings and electrode series resistance compensated using bridge balance in current-clamp mode. Recordings with series resistance > 20 MΩ were discontinued. Constant current injection maintained the membrane potential of pyramidal neurons during whole cell recording at ~−65 mV. Action potentials trains were initiated by somatic current injection (300 ms). To measure passive parameters in each recording, a −20 pA current step was utilized. For analysis of all passive and active parameters, Clampfit software (Molecular Devices) and custom python scripts were utilized.

## Data availability

The mass spectrometry proteomics data have been deposited to the ProteomeXchange Consortium via the PRIDE partner repository with the dataset identifiers PXD027488 and PXD032161. The 2020 mouse Uniprot database (downloaded from https://www.uniprot.org/help/reference_proteome) was used for searches of mass spectrometry data. Source data are provided with this paper.

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

## Acknowledgements

Research reported in this publication was supported by the National Institute on Aging of the National Institutes of Health and National Institutes of Health: F32AG064862 (S.Rayaprolu), R01 NS114130 (SR), K08 NS099474 (SR), RF1 AG071587 (SR, NTS), R01 AG075820 (SR. NTS, LBW), R01AG061800 (NTS), RF1AG062181 (NTS), U01AG061357 (AIL, NTS), U54AG065187 (PI: AIL; RB), 1R01NS115994 (LBW), NSF CAREER 1944053 (LBW), 1 R56 AG072473-01 (MR); Goizueta Alzheimer's Disease Research Center: P30 AG066511 (PI: AIL; RB), 00100569 (MR); P50 AG025688 (PI: AIL - Pilot award to SR and LBW). We acknowledge Karolina Piotrowska-Nitsche, PhD for contributions via the Emory Transgenic Core, and Dr Tamary Caspary and Rachel Bear for donation of a breeding pair of Aldh1l1-Cre-Ert2 mice. Research reported in this publication was also supported in part by the Emory Integrated Proteomics Core (EIPC), Emory Integrate Genomic Core (EIGC), Mouse Transgenic and Gene Targeting Core (TMF), and Emory University Integrated Cellular Imaging (ICI) Microscopy Core which are subsidized by the Emory University School of Medicine and is one of the Emory Integrated Core Facilities. Additional support was provided by the Georgia Clinical & Translational Science Alliance of the National Institutes of Health under Award Number UL1TR002378 (EIPC & EIGC). Additional support to TMF was provided by the National Center for Advancing Translational Sciences of the National Institutes of Health under Award Number UL1TR000454.

## Author contributions

Conceptualization: S.Rayaprolu, N.T.S., S.Rangaraju, Methodology: S.Rayaprolu, S.B., R.B., S.S., L.C., H.X., J.V.S., P.K., P.B., D.M.D., R.N., A.M.G., V.J.O., L.B.W., N.T.S., S.Rangaraju Investigation: S.Rayaprolu, S.B., R.B., S.S., L.C., H.X., P.B., D.M.D., R.N., A.M.G., V.J.O., N.T.S., S.Rangaraju. Writing-Original draft: S.Rayaprolu, N.T.S., S.Rangaraju, Writing-Review and Editing: S.Rayaprolu, N.T.S., S.Rangaraju, S.S., R.B., S.B., P.B., D.M.D., J.V.S., P.K., M.R., A.I.L., L.B.W. Funding acquisition: S.Rayaprolu, S.Rangaraju, N.T.S., A.I.L., L.B.W. Resources: S.Rangaraju, N.T.S., A.I.L., M.R. Supervision: S.Rangaraju, N.T.S.

## Competing interests

The authors declare no competing interests.
