## [Peer Review File · Nature Communications]

REVIEWER COMMENTS

Reviewer #1 (Remarks to the Author):

Rayaprolu and colleagues present methods for performing cell-type specific proteome profiling utilizing Cre-dependent biotin ligase expression in mouse brain, activated using Cre-expressing viruses and transgenic mice. In addition to a technical demonstration the authors present evidence for brain regional differences in expression of CamK2a neurons. Using AAV9-HSyn-Cre the authors demonstrate the feasibility of recovering 2,000+ proteins and the authors perform bioinformatic analysis, which describes the set of proteins; the results are unsurprising but a nice demonstration of feasibility. The transgenic mouse approach using CamK2a-Cre/ERT2 is nicely performed and with controls and identified regional enrichments; the results are unsurprising but a nice demonstration of feasibility. The Luminex studies show further feasibility.

Overall, this is a very thorough paper, makes a solid contribution to the problems of cell-type specific proteomics and presents useful new tools and datasets. This paper is really a technical demonstration paper and does not advance any new scientific concepts or shed significant light on biological questions. The disease enrichment studies and descriptions of what certain genes are known to do doesn't really add anything substantive to the paper, but is useful as a demonstration of a framework that they promote in the conclusions

Minor:

Cell-type specific proteomic studies of synapses using Cre-loxp methods has been performed previously and it would be appropriate to refer to this work in the introduction (Zhu et al, European Journal of Neuroscience, 2019. <https://doi.org/10.1111/ejn.14597>).

Brain regional synapse proteomics studies have been previously published in mice (Roy et al, Proteomes 2018, 6(3), 31; <https://doi.org/10.3390/proteomes6030031>) and humans (Roy, M., Sorokina, O., Skene, N. et al. Proteomic analysis of postsynaptic proteins in regions of the human neocortex (Nat Neurosci 21, 130–138 (2018). <https://doi.org/10.1038/s41593-017-0025-9>) and could be referenced.

Reviewer #2 (Remarks to the Author):

This manuscript describes a method adaptation/development study where a mouse line has been generated that enables cell type-specific expression of the biotin ligase, TurboID, via a Cre-lox strategy. In conjunction with the use of adenoviral and transgenic approaches, this enables in vivo proximity-dependent biotinylation of proteins followed by affinity purification and LC-MS/MS proteomics. The work focusses on neurons in mouse brain, where more than 2,000 neuron-derived proteins were potentially identified and quantitated. Using an inducible CaMKIIA-Cre line, specific proteomic profiles of CaMKIIa-expressing neuronal populations in different brain regions are compared. Overall the study is technically solid, with a possible caveat related to how missing values in the MS data are treated. Other than general confirmation of neuronal protein identity in subsets of neuronal populations, there is little novel biological insight provided.

Specific comments:

1. Abstract – L36 – “Remarkably” seems like an over-statement!
2. Abstract – As written, the phospho-peptide and cytokine work seems disconnected from a cell type workflow – explain better in Abstract.
3. L575- biotin supplementation – provide details.
4. L688 – as written, the use of a filter to allow for up to 6 of 9 missing values seems liberal. How many proteins can be confidently identified using a more stringent filter.
5. L184/L696 – how important is level of TurboID expression? – is there a linear relationship between protein biotinylation and/or suppression of endogenous biotinylation?
6. L233/l284 – striatum/thalamus – only 6 proteins identified – maybe more would have been found if striatum had been analyzed independently.
7. Fig 4a – could presence of crosslinked biotin interfere with detection Ab in standard assay?
8. Explain better the make-up of the Millipore capture Ab panels – presumably these are phospho-site specific.
9. L311 – Fig 4c,d,e for standard and adapted assay data provide some sort of statistical analysis.

Reviewer #3 (Remarks to the Author):

This manuscript from Rayaprolu et al. describes a new conditional TurboID mouse model that enables cell type-specific proteomic analysis of mouse brain. Experiments are shown to validate the conditional mouse and proteome labeling strategy. The authors then use this model to establish proteome, phosphoproteome, and cytokine profile from a genetically defined neuronal population (Camk2a+) across five brain regions. Compared to the established literature, this work is highly complementary to the published BONCAT and APEX work (PMIDs 29106408 and 34381044, respectively). Importantly, this conditional TurboID mouse model is an important addition to the available tools for cell type-specific proteome analysis in an intact animal. In my opinion there would be considerable interest in this conditional mouse model in the neuroscience and related communities. However, I also have some moderate to major concerns regarding the current set of experiments as described below.

1. The authors claim that their mouse model enables cell type-specific proteomics in mouse brain, but this claim is only demonstrated for one cell type. The inclusion of at least one other cell type proteome would be required to truly demonstrate cell type specificity, establish the generality across multiple cell types, and to provide datasets for comparing proteomes between two cell types (e.g., excitatory versus inhibitory neurons, or neurons vs. glia, etc.)

2. While the proteomics data collection are performed to a high standard, I have a concern about the data analysis and interpretation of brain-region specific proteomes (Fig 3) which would require revision. The authors do not control for potential region-specific differences in background proteomes. The PCA shown in Fig 3b might simply be due to basal differences in protein expression in the different regions. In fact, manual inspection of the raw proteomics data in Supp Table 10 already identifies several examples of this (e.g., enrichment of CALB1 in total cerebellum proteome; enrichment of LINGO1 in total hippocampus and cortex proteome). The region-specific analysis should include controls for background proteome differences between brain regions. Alternatively, inclusion of a second cell type proteome across these five regions would also help to provide a background proteome in those brain regions.

3. I found Fig 4 to be confusing and lacking appropriate controls. First, for Fig. 4a-e, no data is shown for a similar assay from WT brain and consequently it is unclear to me whether the signals correspond with true signals from TurboID-labeled proteins or whether they are simply background signals from the total brain. Furthermore, the appropriate control to establish a neuronal or non-neuronal source for these detected proteins should be included (e.g., a glial or mural cell driven TurboID control). The authors do not describe by what statistical or numerical criteria were used to determine whether specific proteins were of neuronal versus non-neuronal origin (e.g., the relative abundance in the adapted assay versus standard assay?). Finally, my same concern about using the

appropriate region-specific background proteomes above also applies to region specific cytokine and phosphoproteome analysis presented in Fig. 4f-h.

4. No description is provided for how the statistics were performed in the supplemental tables. My manual t-test calculations from Excel differs from those values reported in the tables. This discrepancy should be clarified.

REVIEWER COMMENTS

Reviewer #1 (Remarks to the Author):

Rayaprolu and colleagues present methods for performing cell-type specific proteome profiling utilizing Cre-dependent biotin ligase expression in mouse brain, activated using Cre-expressing viruses and transgenic mice. In addition to a technical demonstration the authors present evidence for brain regional differences in expression of CamK2a neurons. Using AAV9-HSyn-Cre the authors demonstrate the feasibility of recovering 2,000+ proteins and the authors perform bioinformatic analysis, which describes the set of proteins; the results are unsurprising but a nice demonstration of feasibility. The transgenic mouse approach using CamK2a-Cre/ERT2 is nicely performed and with controls and identified regional enrichments; the results are unsurprising but a nice demonstration of feasibility. The Luminex studies show further feasibility. Overall, this is a very thorough paper, makes a solid contribution to the problems of cell-type specific proteomics and presents useful new tools and datasets. This paper is really a technical demonstration paper and does not advance any new scientific concepts or shed significant light on biological questions. The disease enrichment studies and descriptions of what certain genes are known to do doesn't really add anything substantive to the paper, but is useful as a demonstration of a framework that they promote in the conclusions.

Response: We thank the reviewer for the comments. We agree that that primary focus of the paper is to demonstrate the technical feasibility and validity of a novel *in-vivo* approach for cell type specific proteomics. While the enrichment of neuron-specific proteins by neuronal CIBOP using both AAV and transgenic strategies is not unexpected, it is a validation of the approach. Using this approach, we identified novel regional differences within one class of neurons. Our additional data now extends CIBOP to label and contrast astrocytes against neurons, providing robust reference proteomic datasets for two common brain cell types. We also included spinal cord as an additional regional contrast in our paper to enhance the novelty of our findings. A major advantage of CIBOP is the ability to characterize labeled cells in their native environment without the artefacts of isolation. We have now compared our Camk2a-CIBOP and Aldh111-CIBOP proteomes and found that the Camk2a-CIBOP reflected synaptic and neuronal functions while Aldh111-CIBOP reflected metabolic and glial functions, confirming that we indeed captured two distinct proteomes with characteristic molecular features.

Minor:

Cell-type specific proteomic studies of synapses using Cre-loxp methods has been performed previously and it would be appropriate to refer to this work in the introduction (Zhu et al, European Journal of Neuroscience, 2019. <https://doi.org/10.1111/ejn.14597>).

Response: We have included this important citation in the introduction.

Brain regional synapse proteomics studies have been previously published in mice (Roy et al, Proteomes 2018, 6(3), 31; <https://doi.org/10.3390/proteomes6030031>) and humans (Roy, M., Sorokina, O., Skene, N. et al. Proteomic analysis of postsynaptic proteins in regions of the human neocortex (Nat Neurosci 21, 130–138 (2018). <https://doi.org/10.1038/s41593-017-0025-9>) and could be referenced.

Response: We have included these references in the introduction and discussion sections and have modified the discussion to contextualize our findings with existing literature.

Reviewer #2 (Remarks to the Author):

This manuscript describes a method adaptation/development study where a mouse line has been generated that enables cell type-specific expression of the biotin ligase, TurboID, via a Cre-lox strategy. In conjunction with the use of adenoviral and transgenic approaches, this enables in vivo proximity-dependent biotinylation of proteins followed by affinity purification and LC-MS/MS proteomics. The work focusses on neurons in mouse brain, where more than 2,000 neuron-derived proteins were potentially identified and quantitated. Using an inducible CaMKIIA-Cre line, specific proteomic profiles of CaMKIIa-expressing neuronal populations in different brain regions are compared. Overall the study is technically solid, with a possible caveat related to how missing values in the MS data are treated. Other than general confirmation of neuronal protein identity in subsets of neuronal populations, there is little novel biological insight provided.

Response: We thank the reviewer for the comments. As discussed above (response to reviewer 1 and response to the Editor), we have attempted to highlight novel biological findings in the discussion. Addition of the second cell type (astrocytes) provides new biological regional differences in astrocyte proteomes.

Specific comments:

1. Abstract – L36 – “Remarkably” seems like an over-statement!

Response: This has been corrected and de-emphasized.

2. Abstract – As written, the phospho-peptide and cytokine work seems disconnected from a cell type workflow – explain better in Abstract.

Response: We have attempted to rephrase the abstract to better integrate the signaling and cytokine work.

3. L575- biotin supplementation – provide details.

Response: We have now added these details in the methods.

4. L688 – as written, the use of a filter to allow for up to 6 of 9 missing values seems liberal. How many proteins can be confidently identified using a more stringent filter.

Response: This comment relates to the AAV-CIBOP experiment which included $n = 3$ WT controls, $n = 3$ hSyn-AAV-Cre/WT controls, and $n = 3$ hSyn-AAV-Cre/*Rosa26^{TurboID/wt}* mice. Of these 9, only 3 mice should show neuronal proteomic labeling. The rationale for using a liberal missingness filter of 6, rather than a traditionally used 50% threshold, was to allow for detection of proteins that showed clear all-or-none expression patterns across groups. We clarify that

using this approach, 0 proteins showed missing values in the positively-labeled mice (hSyn-AAV-Cre/*Rosa26^{TurboID/wt}*). Several proteins exhibited all-or-none expression patterns, which would be otherwise excluded if more stringent filters were applied. The number of total identified proteins increases ($N = 552, 894, 1350, 1831, 2144, 2241, 2258$) as the missingness filter is changed from 6 to 0. A traditional 50% missingness threshold (missing in >4 out of 9 samples) only excludes an additional 114 proteins from the final analysis. Interestingly, 113 of these 114 proteins were also differentially expressed with higher levels in the labeled mice (0 missing values in the labeled mice); and enrichment analysis of these 113 proteins (e.g., *Gabbr2*, *Gabrg1*, *Slc30a1*, *Kcnd3* and many others) confirms enrichment of neuronal/synaptic proteins (**Reviewer Figure 1**), emphasizing that these proteins still represent neuron-derived proteins. Based on this argument, we hope to justify retaining the current filtering approach.

5. L184/L696 – how important is level of TurboID expression? – is there a linear relationship between protein biotinylation and/or suppression of endogenous biotinylation?

Response: Yes, there is indeed a linear relationship between TurboID expression and level of biotinylation. Quantitation of western blots (V5 indicating TurboID levels) and total biotinylation shows this clear relationship (**Reviewer Figure 2**). In the biotin-enriched proteomes, we also find a clear linear relationship between TurboID abundance and sum intensity values. For both the above these reasons, we normalized our proteomic data to TurboID abundance for that given sample as measured by MS, so that any observed regional differences are not related to efficiency of TurboID expression or efficiency of biotinylation, but rather true regional biological differences in the intrinsic proteomes of labeled cell types.

6. L233/I284 – striatum/thalamus – only 6 proteins identified – maybe more would have been found if striatum had been analyzed independently.

Response: We agree that our failure to separate the striatum and thalamus into discrete regions prevented us from appreciating differences in these regions. Since our first dataset used this dissection strategy, we continued to use this approach for subsequent *in vivo* studies related to this manuscript to maintain consistency. Our future studies will undertake a more detailed regional dissection approach.

7. Fig 4a – could presence of crosslinked biotin interfere with detection Ab in standard assay?

Response: Yes, we agree with the reviewer. Biotinylation of proteins of interest (signaling proteins or cytokines) in the labeled mice should result in an inflation in the Luminex readout using the standard assay approach. This is because the pre-biotinylated proteins will directly

bind to detection antibody in addition to binding to the capture antibody, resulting in a higher total signal. Consistent with this, we observed slightly higher signals using the standard assay, when comparing labeled mouse brain to non-labeled mouse brains. This difference (delta) should theoretically correlate with the readout from the adapted assay, provided the labeling strategy itself does not induce changes in neurons. While the interpretation of standard assay results from the AAV-CIBOP approach can be partly confounded by the effects of AAV delivery, the transgenic CIBOP approach would be mostly devoid of this. To address the reviewer's concern, we first compared standard assay results from controls and labeled mice from the transgenic Camk2a-CIBOP study. Of the 53 analytes measured, 48 exhibited this delta (>0) in the standard assay and among these analytes, the delta in the standard assay was significantly and moderately correlated (Pearson R = 0.77, $p < 0.001$) with the Luminex readouts using the adapted assay (**Reviewer Figure 3**). Only 1 protein (phospho-PTEN) showed a non-significant negative delta value. Overall, these analyses suggest that any interference due to pre-biotinylation (e.g., cross-linking of biotin or assay interference) should result in an inflated signal in the standard assay, and that this inflation reflects the biotinylation of the specific analyte in the labeled cell type.

We also directly determined whether the signal detected in the adapted assay was truly from biotinylated protein rather than some other artifact, we added a blocking step before sample addition (addition of 0.05% avidin, followed by 0.05% biotin) to quench the signal derived from proteins already biotinylated by TurboID. This successfully abolished the adapted assay signal across all analytes from the Camk2a-CIBOP cohort, across two Luminex panels (Akt/mTOR and cytokines) verifying the specificity of the adapted assay. We have included these new data in **Supplemental Fig 8** (Also shown in this response).

8. Explain better the make-up of the Millipore capture Ab panels – presumably these are phospho-site specific.

Response: We have included these details in the Methods section. All the antibodies for signaling phospho-proteins are phospho-site specific.

9. L311 – Fig 4c,d,e for standard and adapted assay data provide some sort of statistical analysis.

Response: Statistical pairwise comparisons for Figure 4 c,d,e are detailed in Supplemental data 17, and we have now indicated this in the legend.

Reviewer #3 (Remarks to the Author):

This manuscript from Rayaprolu et al. describes a new conditional TurboID mouse model that enables cell type-specific proteomic analysis of mouse brain. Experiments are shown to validate the conditional mouse and proteome labeling strategy. The authors then use this model to establish proteome, phosphoproteome, and cytokine profile from a genetically defined neuronal population (Camk2a+) across five brain regions. Compared to the established literature, this work is highly complementary to the published BONCAT and APEX work (PMIDs 29106408 and 34381044, respectively). Importantly, this conditional TurboID mouse model is an important addition to the available tools for cell type-specific proteome analysis in an intact animal. In my opinion there would be considerable interest in this conditional mouse model in the neuroscience and related communities. However, I also have some moderate to major concerns regarding the current set of experiments as described below.

Response: We sincerely appreciate the reviewer's comments

1. The authors claim that their mouse model enables cell type-specific proteomics in mouse brain, but this claim is only demonstrated for one cell type. The inclusion of at least one other cell type proteome would be required to truly demonstrate cell type specificity, establish the generality across multiple cell types, and to provide datasets for comparing proteomes between two cell types (e.g., excitatory versus inhibitory neurons, or neurons vs. glia, etc.)

Response: We have now included a second cell type (astrocytes via Aldh111-Cre^{ERT2} mice) to contrast with Camk2a neurons using the transgenic CIBOP approach. These new findings have been included in Figure 5 with additional supplemental figures and supplemental data tables. We show that the Camk2a neuronal proteome is different from the Aldh111 astrocyte proteome, further supported by immunohistochemical studies. Based on these new data, we effectively demonstrate the ability of our mouse model to resolve *in vivo* native-state proteomes of two different cell types in the brain, with the potential to extend to other tissue/cell type systems using existing Cre genetic models.

2. While the proteomics data collection are performed to a high standard, I have a concern about the data analysis and interpretation of brain-region specific proteomes (Fig 3) which would require revision. The authors do not control for potential region-specific differences in background proteomes. The PCA shown in Fig 3b might simply be due to basal differences in protein expression in the different regions. In fact, manual inspection of the raw proteomics data in Supp Table 10 already identifies several examples of this (e.g., enrichment of CALB1 in total cerebellum proteome; enrichment of LINGO1 in total hippocampus and cortex proteome). The region-specific analysis should include controls for background proteome differences between brain regions. Alternatively, inclusion of a second cell type proteome across these five regions would also help to provide a background proteome in those brain regions.

Response: We have now included data from background regional proteomes from the Camk2a-CIBOP study in Supplemental data 16, related to Figure 3. 3,969 proteins were identified in the background proteome, of which 1872 were also identified in the Camk2a-CIBOP proteome. 201 proteins were exclusively identified in the Camk2a-CIBOP proteome. Within these 1872 proteins, we identified proteins that demonstrated regional enrichment within the

background or within the Camk2a-CIBOP proteomes separately, and then assessed the degree of overlap between regionally enriched proteins. In the hippocampus, Camk2a-CIBOP identified 549 regionally enriched proteins of which 419 (76.3%), including Lingo1, were also regionally enriched in the background proteome, likely due to high proportion of Camk2a neurons in these regions (over 50% of all cells). In contrast, lower levels of overlap were observed in other regions (9.6% in striatum/thalamus, 58% in the pons/medulla and 59% in the cerebellum). Using definitions of core regional protein markers (≥ 4 -fold enrichment in the specific region over other regions and $p < 0.05$), 1,110 proteins were identified as core-markers in the Camk2a-CIBOP proteome, of which only 343 were also identified as core markers in the background proteome. This shows that the Camk2a-CIBOP proteome identifies twice as many core regional proteomic differences within Camk2a neurons that were not captured by the background proteome. These data and analyses are shown in **Supplemental Fig 7**.

Another limitation is that Camk2a is not entirely specific to excitatory neurons. Rather, Camk2a expression has been demonstrated in several neuronal classes across the brain in mice. Therefore, future CIBOP studies with Cre driver lines that are more specific to neuronal subtypes, will be very informative. We have also included a second cell type across all regions to directly contrast against neurons (shown in Figure 5).

3. (i) I found Fig 4 to be confusing and lacking appropriate controls. First, for Fig. 4a-e, no data is shown for a similar assay from WT brain and consequently it is unclear to me whether the signals correspond with true signals from TurboID-labeled proteins or whether they are simply background signals from the total brain. (ii) Furthermore, the appropriate control to establish a neuronal or non-neuronal source for these detected proteins should be included (e.g., a glial or mural cell driven TurboID control). (iii) The authors do not describe by what statistical or numerical criteria were used to determine whether specific proteins were of neuronal versus non-neuronal origin (e.g., the relative abundance in the adapted assay versus standard assay?). (iv) Finally, my same concern about using the appropriate region-specific background proteomes above also applies to region specific cytokine and phosphoproteome analysis presented in Fig. 4f-h.

Response: We have addressed each point raised in the above comment in an itemized manner.

(i) **Lack of appropriate controls and lack of clarity:** We have now included standard and adapted assay data from AAV-CIBOP and transgenic Camk2a-CIBOP experiments as a supplemental table. Figure 4 has also been modified to improve clarity. Figure 4 panels c-e show standard and adapted assay results from the AAV-CIBOP experiment. The standard assay data shown in the figure is from the AAV-CIBOP labeled brains. The adapted assay data represents the adapted Luminex data after subtracting out signal from control mice, to account for any background non-specific signal. As shown in the supplemental table (and figure alongside for the Reviewer's consideration), adapted assay readouts from hSyn-AAV control samples in the AAV-CIBOP cohort, and from control Camk2a-Cre mice (in the transgenic CIBOP cohort), were very small compared to signals derived from labeled mice. This confirms that the signals being measured by the adapted assay are not due to background or non-specific signals from the total brain. Since all control and positive mice also received biotin in the exact same way, any residual free biotin in the brain could not have contributed to spurious signal either. Furthermore, blocking with avidin before addition of the capture antibody in the adapted assay abolished the signal confirming that the signal being detected was due to biotinylation (**Supplemental Fig 8**).

(ii) **Neuronal vs non-neuronal source:** We agree with the reviewer. Accordingly, we have now included Luminex data focusing on signaling phospho-proteins contrasting Camk2a-CIBOP with astrocyte Aldh1l1-CIBOP. The findings are presented in Figure 5. This cell type contrast shows that signaling phospho-proteins, particularly those in the MAPK pathway (e.g., pERK1/2 and pMEK1/2) are predominantly derived from Camk2a neurons in the adult mouse brain, across regions.

(iii) **Criteria used to determine neuronal versus non-neuronal origin:** We would like to clarify the criteria to classify a measured protein as being primarily neuronal vs non-neuronally derived. If the adapted assay signal (after subtracting control values) was greater than half of the standard signal from the same animal, we classified that protein as being primarily derived from the labeled cell type, in this case: neuronal origin. As requested by Reviewer 1, we have also included statistical comparisons between standard and adapted signals in the AAV-CIBOP study shown in Supplemental Data 17.

(iv) **Region-specific background cytokine and phosphoproteome:** We would like to clarify that Camk2a-CIBOP region-specific cytokine and phosphoproteome analyses presented in Fig. 4f-h were done on *total brain lysates* and not on IP proteins. This is indeed the advantage of using the Luminex assays with our biotinylated proteome – we do not need to IP/pulldown biotinylated proteins in order to detect the cytokines or phosphoproteins; we can directly add the total brain lysates onto the plates to detect the proteins using the adapted assay. The data for these studies are provided in Supplemental Table 20.

To further characterize the data from these studies, we performed combined PCA analysis of standard assay readouts (background signal) and adapted assay readouts (Camk2a-specific signal) from total brain lysates (Reviewer Figure 5). This showed that PC1 (26% of variance) captured regional differences using the adapted assay that were not captured at the background level. PC2 (9% of variance) captured regional differences that were consistent across standard and adapted assays, although the magnitude of difference between regions was still more apparent in the adapted assays as compared to the standard assay. These additional analyses show that majority of regional differences in Camk2a neuron-derived signals (cytokines and phospho-proteins) cannot be explained by underlying regional differences at the background level.

4. No description is provided for how the statistics were performed in the supplemental tables. My manual t-test calculations from Excel differs from those values reported in the tables. This discrepancy should be clarified.

Response: We have included details in the supplemental tables regarding statistical test used for each comparison. We have also verified T test statistics in the supplemental table (2 tailed, assuming equal variance) and corrected any inaccuracies if present.

REVIEWERS' COMMENTS

Reviewer #2 (Remarks to the Author):

The revised version of this manuscript describes a method adaptation/development study where mouse lines have been generated to enable cell type-specific expression of the biotin ligase, TurboID, via an inducible Cre-lox strategy, in brain. In conjunction with the use of adenoviral and transgenic approaches, this enables in vivo proximity-dependent biotinylation of proteins followed by affinity purification and LC-MS/MS proteomics. The original work focused on neurons in mouse brain, where more than 2,000 neuron-derived proteins were potentially identified and quantitated. Using an inducible CaMKIIA-Cre line, specific proteomic profiles of CaMKIIa-expressing neuronal populations in different brain regions were compared. Overall, the study was considered technically solid, but other than general confirmation of neuronal protein identity in subsets of neuronal populations, there was little novel biological insight provided. To address the comments from the previous reviewers, the main revision to the manuscript is the addition of data from a new mouse line that expresses TurboID in astrocytes, enabling regional comparison of astrocytic proteomes, and comparison of astrocytic and CaMKIIa neuron proteomes. While the new data helps address feasibility of the approach, this does little to add in terms of new biological insight.

Specific comments:

1. Use of NES construct – presumably this explains the lack of coverage of nuclear proteins?
2. L148 – 7 proteins noted in text vs 5 indicated in Fig 1f?
3. L307 – “as well as”
4. L366 - “Remarkedly” seems like an over-statement!
5. On re-reading the manuscript, I found the data in Figure 4c,d,e to not be very compelling.
6. L420 – “gliosis in”?
7. Fig 5 – are the CaMKIIa samples different from prior samples in Fig 2/3?
8. Fig 5f/g – there is potential in the new astrocyte data for validation and follow-up functional studies?
9. L450-490 – conclusions are anti-climactic – new data confirm what is known already.
10. L550 and elsewhere – “With this approach, we had the opportunity to characterize regional proteomic differences within the same neuronal class” – The expression of CaMKIIa does not define

a neuronal class, eg there is high expression in striatal medium spiny neurons which are GABAergic projection neurons.

11. L568 – “that is predisposes”

12. L611 – “while synaptic transmission and long-term potentiation were specific to the neuronal proteome” – seems to contradict lines 597/598.

Reviewer #3 (Remarks to the Author):

The authors have adequately addressed my concerns

Date 04/27/2022

We appreciate the overall positive comments on the revised manuscript. We have addressed Reviewer 2's minor comments below. In addition, we have revised the manuscript to meet editorial guidelines and policies of Nature Communications as requested. We thank you for your consideration.

REVIEWERS' COMMENTS

Reviewer #2 (Remarks to the Author):

The revised version of this manuscript describes a method adaptation/development study where mouse lines have been generated to enable cell type-specific expression of the biotin ligase, TurboID, via an inducible Cre-lox strategy, in brain. In conjunction with the use of adenoviral and transgenic approaches, this enables in vivo proximity-dependent biotinylation of proteins followed by affinity purification and LC-MS/MS proteomics. The original work focused on neurons in mouse brain, where more than 2,000 neuron-derived proteins were potentially identified and quantitated. Using an inducible CaMKIIA-Cre line, specific proteomic profiles of CaMKIIa-expressing neuronal populations in different brain regions were compared. Overall, the study was considered technically solid, but other than general confirmation of neuronal protein identity in subsets of neuronal populations, there was little novel biological insight provided. To address the comments from the previous reviewers, the main revision to the manuscript is the addition of data from a new mouse line that expresses TurboID in astrocytes, enabling regional comparison of astrocytic proteomes, and comparison of astrocytic and CaMKIIa neuron proteomes. While the new data helps address feasibility of the approach, this does little to add in terms of new biological insight.

Specific comments:

1. Use of NES construct – presumably this explains the lack of coverage of nuclear proteins?

Response: The reviewer is correct, that the inclusion of the NES sequence by design, is the likely explanation for low coverage of nuclear proteins. This is also supported by predominant biotinylation in the cell bodies of labeled neurons and astrocytes in mouse brain tissues. A statement to this effect has been included in the first paragraph of the discussion section.

2. L148 – 7 proteins noted in text vs 5 indicated in Fig 1f?

Response: This has been corrected to 5. We apologize for the error.

3. L307 – “as well as”

Response: This has been corrected.

4. L366 - “Remarkedly” seems like an over-statement!

Response: We have removed “Remarkedly”

5. On re-reading the manuscript, I found the data in Figure 4c,d,e to not be very compelling.

Response: We have attempted to rephrase the results related to Figures 4c, d and e to make the finding clearer.

6. L420 – “gliosis in”?

Response: This has been corrected.

7. Fig 5 – are the CaMKIIa samples different from prior samples in Fig 2/3?

Response: An independent set of CaMKIIa-CIBOP mice were used in experiments related to Figure 5, along with the new astrocyte lines, in order to ensure a fair comparison. This has been shown in the cartoon in Figure 5 panel. Mice related to Fig 2 and 3 were NOT used in studies in Fig 5. We have further clarified this in the results section related to Fig 5.

8. Fig 5f/g – there is potential in the new astrocyte data for validation and follow-up functional studies?

Response: Yes, we agree that demonstrating the feasibility of labeling and capturing astrocytic proteomes *in vivo* lays the foundation for several validation and follow-up functional studies, which can be undertaken on a protein-by-protein basis, which we felt was beyond the scope of the current work.

9. L450-490 – conclusions are anti-climactic – new data confirm what is known already.

Response: The primary goal of this manuscript is to highlight the development, validation and applicability of this model and the CIBOP approach for cell type specific proteomics. While finding known neuronal or known astrocyte proteins in the respective cell type is not necessarily novel, it is important to verify these to ensure validity of the model. In

addition to this, we identified several proteins by neuronal CIBOP and astrocyte CIBOP which were otherwise not identified in prior cell type specific proteomes, nor in the bulk brain proteomes. Other conclusions related to regional differences within the same cell type, and the differences in MAPK and Akt/mTOR signaling patterns between these two cells are also relatively novel. While new disease-relevant insight cannot be gained by our currently reported studies, we lay the foundation for the next series of experiments in disease models, to investigate how cellular proteomic profiles change *in vivo*, in their native state.

10. L550 and elsewhere – “With this approach, we had the opportunity to characterize regional proteomic differences within the same neuronal class” – The expression of CaMKIIa does not define a neuronal class, eg there is high expression in striatal medium spiny neurons which are GABAergic projection neurons.

Response: We agree with the reviewer. While Camk2a is highly expressed by excitatory neurons, several other neuronal classes including GABAergic projection neurons also express CAMKIIa. Therefore, we have edited these statements throughout out the manuscript and have refrained from using the term ‘class’ or ‘excitatory neurons’ when referring to Camk2a neurons in this paper.

11. L568 – “that is predisposes”

Response: This has been corrected.

12. L611 – “while synaptic transmission and long-term potentiation were specific to the neuronal proteome” – seems to contradict lines 597/598.

Response: We apologize for this apparent contradiction. Proteins involved in synaptic function were enriched in the astrocyte proteome, as compared to biotin pulldowns from unlabeled mice. This means these proteins were present and labeled in astrocytes. We have commented on potential explanations for detection of synaptic proteins in astrocytes. When astrocyte and neuronal CIBOP proteomes are contrasted, synaptic proteins are enriched in the neuronal CIBOP proteomes. Rather than ‘specific’ we have now used the term ‘enriched’ and clarified the contrast between neurons and astrocytes in L611.

Reviewer #3 (Remarks to the Author):

The authors have adequately addressed my concerns

Response: No concerns to address.